# Dual-frame Fluid Motion Estimation with Test-time Optimization and Zero-divergence Loss

**Yifei Zhang**
University of Chinese Academy of Sciences
zhangyifei21a@mails.ucas.ac.cn

**Huan-ang Gao**
AIR, Tsinghua University
gha24@mails.tsinghua.edu.cn

**Zhou Jiang**
Beijing Institute of Technology
jzian@bit.edu.cn

**Hao Zhao**
AIR, Tsinghua University
Beijing Academy of Artificial Intelligence
zhaohao@air.tsinghua.edu.cn

## Abstract

3D particle tracking velocimetry (PTV) is a key technique for analyzing turbulent flow, one of the most challenging computational problems of our century. At the core of 3D PTV is the dual-frame fluid motion estimation algorithm, which tracks particles across two consecutive frames. Recently, deep learning-based methods have achieved impressive accuracy in dual-frame fluid motion estimation; however, they exploit a supervised scheme that heavily depends on large volumes of labeled data. In this paper, we introduce a new method that is **completely self-supervised and notably outperforms its supervised counterparts while requiring only 1% of the training samples (without labels) used by previous methods.** Our method features a novel zero-divergence loss that is specific to the domain of turbulent flow. Inspired by the success of splat operation in high-dimensional filtering and random fields, we propose a splat-based implementation for this loss which is both efficient and effective. The self-supervised nature of our method naturally supports test-time optimization, leading to the development of a tailored Dynamic Velocimetry Enhancer (DVE) module. We demonstrate that strong cross-domain robustness is achieved through test-time optimization on unseen leave-one-out synthetic domains and real physical/biological domains. Code, data and models are available at https://github.com/Forrest-110/FluidMotionNet.

## 1 Introduction

Measuring and understanding turbulent fluid flow is a crucial problem as it is ubiquitous in various aspects of our lives, both in nature [27; 37; 20] and within our engineered society [28; 81; 35; 26; 69; 101; 68; 93]. Throughout history, flow visualization techniques have played a vital role in quantifying and analyzing turbulent flow [4; 60; 16; 76]. Among existing flow visualization techniques, 3D particle tracking velocimetry (3D PTV), which tracks individual particles between consecutive frames, distinguishes itself with high spatial resolution and precise measurement of velocity vectors [32; 3]. Before the advent of deep learning, PTV algorithms primarily focus on designing hand-crafted features for particle matching [62; 12; 99]. With the onset of deep learning, deep neural networks (like DeepPTV [50] and GotFlow3D [53]) have been introduced to solve this task. The core algorithm of 3D PTV is dual-frame fluid motion estimation, as illustrated in Fig. 1.

However, it is important to note that existing state-of-the-art (SOTA) dual-frame fluid motion estimation algorithms (shown in left two panels of Fig. 1) have a limitation: they require a large amount of fully annotated data. It is known that deep learning generally requires a substantial amount of

38th Conference on Neural Information Processing Systems (NeurIPS 2024).

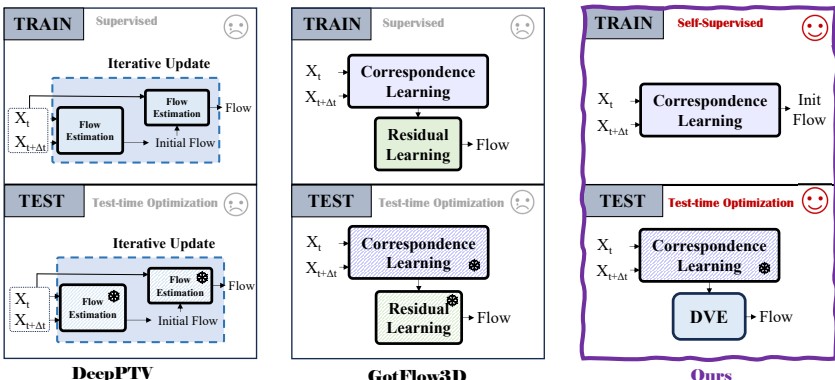

Figure 1: **Paradigm Shift**: Given two frames of flow particles $X_t$ and $X_{t+\Delta t}$, DeepPTV [50] adopts a two-stage network for large- and small-scale motion refinement. GotFlow3D [53] trains a correspondence learning network and an RNN-based residual prediction network. They are trained in a fully supervised manner with annotated data and do not support test-time optimization. Our purely self-supervised method diverges from these approaches and employs DVE (see Sec. 3.3) for on-the-fly test-time optimization. The "Snowflake" denotes frozen weights.

in-domain data for optimal results. This demand poses a significant challenge to the AI4Science field, especially in 3D PTV where collecting suitable data is complicated due to the need for precisely selected tracer particles, tailored illumination, and camera settings [63]. Additionally, certain scenarios like flow fields under unique geometric conditions or cytoplasmic flows in disease contexts are rare, making it nearly impossible to compile a comprehensive dataset.

To alleviate the aforementioned challenge, in this paper, we introduce a novel purely **self-supervised** framework with **test-time optimization** designed specifically for dual-frame fluid motion estimation in the 3D PTV process, as highlighted in the right panel of Fig. 1. Concerning the intrinsic difficulties associated with PTV data collection, especially in specialized contexts [6; 74; 64], we consider working under a limited size of dataset, as little as 1% typically used by existing fully-supervised methods (**notably without accessing labels**). Fluid particles have special physical properties, for which we resort to the inherent zero-divergence principle of incompressible fluid velocity fields and design a novel zero-divergence self-supervised loss tailored for fluid. Per implementation, we introduce the successful idea of splat in high-dimensional filtering [2] and random fields [38] and design a splat-based zero-divergence loss that is both efficient and effective.

Moreover, since our method is self-supervised, it naturally supports test-time optimization. Thus we introduce a module termed *Dynamic Velocimetry Enhancer* (DVE), shown in the right panel of Fig. 1, which optimizes the initial predicted flow during test-time based on the specific input data on the fly, ensuring an improved level of accuracy across various testing scenarios. This is critical for **cross-domain robustness**. The difficulty in collecting diverse PTV data leads to the common practice of using synthetic datasets. However, since synthetic data is generated based on hand-crafted priors, it cannot accurately represent specific real-world distributions, resulting in models that lack the necessary cross-domain robustness for practical applications.

Through comprehensive experiments, we demonstrate that our purely self-supervised framework (right panel of Fig. 1) significantly outperforms its fully-supervised counterparts (left two panels of Fig. 1), even under data-constrained conditions (using as low as 1% data). Additionally, our cross-domain robustness analyses confirm the framework's intrinsic ability to generalize to unseen domains, including leave-one-out synthetic domains and real-world physical/biological domains, underscoring the practical utility of our approach for real-world 3D PTV applications.

To summarize, our main contributions are: 1. A novel self-supervised framework with test-time optimization for dual-frame fluid motion estimation, surpassing fully-supervised methods with minimal samples (as low as 1%). 2. A splat-based zero-divergence self-supervised loss for fluid dynamics, which is both efficient and effective. 3. A test-time optimization module named Dynamic Velocimetry Enhancer (DVE) that significantly improves cross-domain robustness.

## 2    Related Work

### 2.1    Test-time Optimization and Test-time Domain Adaptation

Test-time optimization, also known as test-time refinement (TTR), exploits the inherent structure of data in a self-supervised manner without requiring ground truth labels [14; 80; 24; 21]. Applications of TTR include point cloud registration [25], depth estimation [7; 9], object recognition [80; 89], human motion capture [86], and segmentation with user feedback [73; 77; 31]. Test-time domain adaptation (TTA), a specific form of TTR, adapts a model trained on a source domain to a new target domain using an unsupervised loss function based on the target distribution [90; 54; 104]. One significant challenge in TTR is achieving per-sample adaptation at test time without compromising inference efficiency. Recent studies [85; 67] have explored using generative models to enable efficient test-time adaptation. In this work, we introduce our DVE module (Sec. 3.3), which conducts test-time optimization but maintains efficiency when compared with prior methods.

### 2.2    Learning-based Scene Flow Estimation

We include this section because our research is closely related to point-based, learning-driven scene flow estimation from point clouds—a key component in understanding scenes through point clouds [8; 84; 43; 22]. Both areas of study concentrate on learning flows or correspondences from two frames of data [97; 102; 103; 100; 55]. Advances in scene flow estimation have been driven by benchmarks such as KITTI Scene Flow [59] and FlyingThings3D [58]. Drawing from the related field of optical flow [15; 30; 79; 82], recent developments in scene flow estimation utilize methods including encoder-decoder architectures [23; 57], multi-scale representations [11; 44; 94], recurrent modules [36; 83; 92], and other strategies [42; 70].

**Self-supervision and Test-time Optimization for Scene Flow.** Self-supervised learning has received attention for scene flow estimation from point cloud data [95; 61; 5; 40; 46; 75; 45] and monocular images [29; 10; 105]. PointPwcNet [95] introduces cycle consistency loss, inspiring Mittal et al. [61] to incorporate it with nearest neighbor loss for establishing point cloud correspondence. This method also employs Chamfer Distance [19], smoothness constraints, and Laplacian regularization for self-supervision. SLIM [5] addresses self-supervised scene flow estimation and motion segmentation simultaneously. Flowstep3d [36] uses a soft point matching module for pairwise point correspondence. Self-supervision naturally supports test-time optimization. Pontes et al. [66] eschew model training for real-time optimization by minimizing the graph Laplacian over source points to enforce rigid flow. Li et al. [48] replace the explicit graph with a neural prior using a coordinate-based MLP to implicitly regularize the flow field. SCOOP [40] combines pre-training on a subset of data to learn soft correspondences and secures initial flows with optimization-based refinement steps.

Our work is distinct from these scene flow methods as our data source is a specific domain: flow particles. Fluid particles differ from typical scene flow point clouds due to their disordered local distribution [51] (see Sec. 3.2.1) and unique physical properties. We propose a graph-based feature extractor and zero-divergence regularization to leverage these properties (see Sec. 3.2).

### 2.3    Particle Tracking Velocimetry (PTV)

Particle tracking is a fundamental tool in turbulence analysis, progressing from traditional methods like streak photography [18] to advanced techniques such as Laser Speckle Photography (LSV) [65]. This evolution establishes the foundation for Particle Tracking Velocimetry (PTV). PTV gains prominence with the development of automatic tracking algorithms [1], which represents a significant advancement over manual methods [13]. Modern PTV calculates velocities by matching particle pairs between frames [1] and has been applied widely across various fields such as materials science, hydrodynamics, biomedical research, and environmental science [28; 35; 101; 27].

**Deep Learning Methods of Dual-frame Fluid Motion Estimation in PTV.** Before the advent of deep learning, PTV algorithms primarily focused on improving particle matching by considering group particle movement [62], using multiple time step data [12], or conducting spatial area segmentation [99]. With the onset of deep learning, deep neural networks have been designed for particle motion estimation from point cloud pairs [57; 50; 53; 70; 96; 92]. Among them, DeepPTV [50] and GotFlow3D [53] are specifically tailored for fluid flow learning and in a fully supervised manner, as

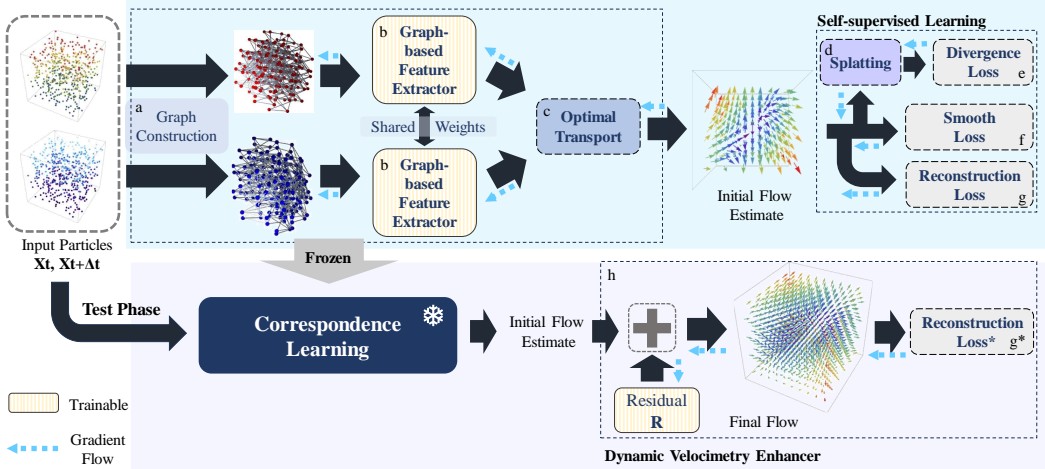

Figure 2: **Upper: Training Phase**. First, we (a) use input point clouds to construct graphs, which are then passed through a trainable (b) feature extractor, and we solve a (c) optimal transport problem using self-supervised loss terms including (g) reconstruction loss, (f) smooth loss, and (e) zero-divergence loss for initial flow estimation. **Lower: (h) Test-Time DVE**. With the initial flow estimate $\mathbf{F}_{\text{init}}$, we optimize a residual $\mathbf{R}$ to generate the final flow $\mathbf{F}$ using another reconstruction loss (g*).

demonstrated in Fig. 1. Our work follows these prior efforts, aiming to develop a data-efficient and cross-domain robust motion estimation technique through self-supervision and test-time optimization.

## 3 Methods

### 3.1 Problem Formulation

To elucidate the architecture and functionality of our proposed method for dual-frame fluid motion estimation, we outline the problem as follows: The method processes two consecutive, unstructured sets of 3D particles, $\mathbf{X_t} \in \mathbb{R}^{n_1 \times 3}$ and $\mathbf{X_{t+\Delta t}} \in \mathbb{R}^{n_2 \times 3}$, recorded at times $\mathbf{t}$ and $\mathbf{t} + \mathbf{\Delta t}$. It outputs the predicted flow motion $\mathbf{F} \in \mathbb{R}^{n_1 \times 3}$, mapping each particle $\mathbf{x}_i$ from $\mathbf{X_t}$ to a vector $\mathbf{f}_i$ that indicates its movement between the two frames, capturing the flow dynamics in the turbulent 3D environment.

### 3.2 Training with Fewer Samples

In the training phase, we aim to learn the patterns of fluid flow using considerably fewer samples, as low as 1% of what conventional approaches require, given the inherent difficulties in gathering data for specific scientific domains. We design the network as depicted at the top of Fig. 2 to train a graph-based feature extractor (Fig. 2b) that extracts per-point features for the following soft point matching. These features initialize the flow between the point clouds using the optimal transport module (Fig. 2c), and we employ self-supervision losses, as shown in Fig. 2(e,f,g), for training. However, fluid particles exhibit complex motion features compared to typical LiDAR point clouds (Sec. 3.2.1), which complicates feature learning under self-supervision with limited data. Consequently, we employ a strong graph-based feature extractor (Sec. 3.2.2) and propose a novel zero-divergence loss (Sec. 3.2.4.3.1) tailored to address these challenges.

**3.2.1 Complexity of Fluid Flow.** A common assumption in LiDAR scene flow estimation is the smoothness of flow. However, this is not enough for fluid particles due to their unique geometric distribution, as shown in the left of Fig. 3. The fluid velocity field is smooth only at a coarse scale but remains complex at a fine local scale. Therefore, we need a strong relation-based graph feature extractor and more specific regularization to capture the intricate properties of fluid particles.

**3.2.2 Graph-based Feature Extractor.** Point cloud-based extractors, including PointNet [71] and PointNet++ [72], are commonly used in LiDAR scene flow estimation [40]. While these extractors effectively discern broader spatial structures, their capability to grasp intricate local relationships, which is vital for analyzing fluid dynamics, can be inadequate. In contrast, graph-based feature

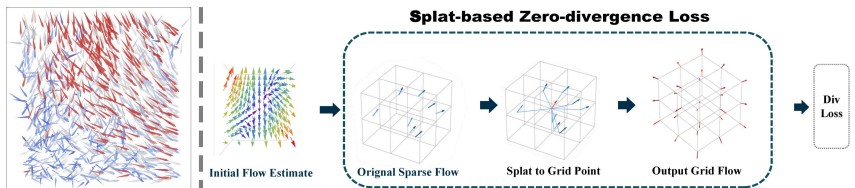

Figure 3: **LEFT:** A visualization of fluid flow in Fluidflow3D data. **RIGHT:** The divergence loss in our training phase is obtained by splatting the original sparse flow to grid points and then minimizing the divergence loss on the resulting grid points.

extractors excel at capturing local patterns by considering the relationships between proximate nodes, or in our context, particles. Hence, drawing inspiration from GotFlow3D [53], we opt for a graph-based feature extraction backbone, as depicted in Fig. 2b. Initially, we construct a static nearest-neighbor graph from the input point cloud. This graph is then processed through several GeoSetConv layers [72] to form a high-dimensional geometric local feature. To further enrich the feature, we construct a dynamic graph using EdgeConv [91] based on the high-dimensional feature, forming a GNN that outputs static-dynamic features. The dynamic graph expands the receptive field and focuses on geometric feature properties. Further details can be found in Appendix A.1.1.

**3.2.3 Solving Optimal Transport for Soft Correspondence.** With the static-dynamic feature from the feature extractor (Sec. 3.2.2), we formulate the correspondence linking problem through the framework of optimal transport [87], where a higher transport cost between two points indicates a lower similarity within the extracted feature space. The optimal transport plan yields the soft correspondence weight between $\mathbf{X_t}$ and $\mathbf{X_{t+\Delta t}}$, as shown in Fig. 2c, which can be used to formulate an initial flow estimate $\mathbf{F}_{\text{init}}$. This follows the common scene flow method, thus we leave the details to Appendix A.2.

**3.2.4 Self-supervised Losses.** Since manually linking particles between sets is notably intricate, we advocate for the adoption of self-supervised losses (Fig. 2e-g).

*3.2.4.1. Reconstruction Loss.* A core principle guiding self-supervised flow learning is the fact that $\mathbf{X_t} + \mathbf{F}$ and $\mathbf{X_{t+\Delta t}}$ should be similar. The Chamfer distance (CD) is a standard metric used to measure the shape dissimilarity between point clouds in point cloud completion. Therefore, we adopt it as our reconstruction loss. We also add a regularization term [40] to prevent degeneration:

$$L_{\text{recon}} = \frac{1}{|\mathcal{Y}'|} \sum_{\mathbf{y}'_i \in \mathcal{Y}'} p_i \min_{\mathbf{y}_j \in \mathcal{Y}} ||\mathbf{y}'_i - \mathbf{y}_j||_2^2 + \lambda_{\text{conf}} \frac{1}{|\mathcal{Y}'|} \sum_{\mathbf{y}'_i \in \mathcal{Y}'} (1 - p_i) \tag{1}$$

Here, $\mathcal{Y}'$ represents the estimated point cloud formed by $\mathbf{X_t} + \mathbf{F}_{\text{init}}$, and $\mathcal{Y}$ is the target point cloud formed by $\mathbf{X_{t+\Delta t}}$. $p_i$ denotes the confidence of matching in the Optimal Transport (Sec. 3.2.3), which is the weighted sum of transport costs. $\lambda_{\text{conf}}$ term is used to avoid the trivial solution $p_i = 0$.

*3.2.4.2. Smooth Loss.* Given the infinitely differentiable characteristic of the velocity field, it is postulated that the field should exhibit a certain level of continuous and smooth transitions (at a coarse scale). In light of this theoretical underpinning, we introduce a smooth regularization loss to enforce and maintain this continuous behavior in the velocity field, which is defined as,

$$L_{\text{smooth}} = \sum_{\mathbf{x}_i \in \mathcal{X}} \sum_{k \in \mathcal{N}_l(\mathbf{x}_i)} \frac{||\mathbf{f}_i - \mathbf{f}_k||_1}{|\mathcal{X}||\mathcal{N}(\mathbf{x}_i)|}, \tag{2}$$

Here, $\mathcal{X}$ represents the point cloud formed by $\mathbf{X_t}$. $\mathcal{N}_l(\mathbf{x}_i)$ represents the index set of the $l$ closest points to $\mathbf{x}_i$. $\mathbf{f}_i$ and $\mathbf{f}_k$ denote the estimated flow vectors at points $\mathbf{x}_i$ and $\mathbf{x}_k$, respectively.

3.2.4.3.1 Zero-divergence Loss. Smooth Loss is not enough for fluid particles, as mentioned in Sec. 3.2.1. Concerning the intrinsic properties of the velocity field, we note that incompressible fluids exhibit zero divergence by definition. Moreover, compressible fluids can also be approximated as incompressible under conditions like low Mach numbers, justifying this in many engineering contexts. Hence, we introduce a zero-divergence regularization, which also compensates for the shortcomings of Smooth Loss, as we will show later.

3.2.4.3.2 Splat-based Implementation. Splatting, first used in high-dimensional Gaussian filtering [2], embeds input values in a high-dimensional space. Studies like [38] and [78] followed this

Splat-Blur-Slice pipeline. Inspired by these, we implemented a splat-based zero-divergence loss: to compute divergence, we need the partial derivative of the field. The irregular arrangement of particles in 3D complicates this. Thus, we propose "splatting" unstructured flow estimates onto a uniform 3D grid, then applying zero-divergence regularization at these grid points, as shown in the right of Fig. 3. In formal terms, the dense grid is denoted by $(sj, sk, sl)^T$, with $j, k, l \in \mathbb{Z}$ indicating the 3D indices of the grid point. The parameter $s$ corresponds to the grid's spacing. Then, given a grid point $\mathbf{x} = (sj, sk, sl)^T$, we employ the inverse squared distance as interpolation weights to approximate the flow at that particular point,

$$\mathbf{f}(\mathbf{x}) = \frac{1}{|N(\mathbf{x})|} \sum_{\mathbf{x}_i \in N(\mathbf{x})} \frac{\mathbf{f}_i}{\|\mathbf{x}_i - \mathbf{x}\|_2^2 + \epsilon} \tag{3}$$

where $\mathbf{f}_i$ is the estimated flow value at point $\mathbf{x}_i$. $N(\mathbf{x})$ denotes the neighborhood among the point set of $\mathbf{X_t}$ for grid point $\mathbf{x}$. The parameter $\epsilon$ is introduced to maintain numerical stability. By employing splatting, we convert the variable particle distance into fixed grid spacing, thus achieving efficiency and effectiveness.

3.2.4.3.3 Divergence Calculation. Once Splatting has been employed, the divergence at that point, specified by $\mathbf{x} = (sj, sk, sl)^T$, can be defined as: $(\nabla \cdot \mathbf{F})(\mathbf{x}) = \sum_{k=1}^{3} \frac{\mathbf{f}(\mathbf{x}+su_k)-\mathbf{f}(\mathbf{x}-su_k)}{2s}$, where $u_k$ is a unit vector with 1 at the $k$-th entry. Finally, the zero-divergence regularization can be formulated as, $L_{\text{div}} = \frac{1}{JKL} \sum_{j=0}^{J-1} \sum_{k=0}^{K-1} \sum_{l=0}^{L-1} \left\| (\nabla \cdot \mathbf{F})((sj, sk, sl)^T) \right\|_1$, where $J$, $K$, and $L$ represent the number of grid points along the respective dimensions.

3.2.4.3.4 Zero-Divergence Loss v.s. Smooth Loss Zero-Divergence loss is similar to Smooth loss in that it computes spatial gradients and requires the norm of the gradient to be small, essentially penalizing the case that neighboring flow vectors are totally irrelevant. However, Smooth regularization is too strict for fluid particles. While the divergence constraint only requires the total divergence to be zero, it does not necessitate that any two vectors be oriented in the same direction, thus allowing for locally complex particle dynamics. In practice, we set the neighborhood set size for calculating Smooth Loss to be much larger than that for calculating Zero-Divergence Loss, because smoothness is a more coarse-scale regularization. Finally, we note that Zero-Divergence Loss is calculated along three specific axes, whereas Smooth Loss is not. Therefore, using the same method (KNN) for calculating Zero-Divergence Loss as for Smooth Loss is not efficient.

To summarize, our final self-supervised training loss is

$$L_{\text{train}} = L_{\text{recon}} + \lambda_{\text{smooth}} L_{\text{smooth}} + \lambda_{\text{div}} L_{\text{div}}$$

### 3.3 Efficient Test-time Optimization with Dynamic Velocimetry Enhancer

As shown at the bottom of Fig. 2, with the initial flow estimate from the trained network, we introduce a novel *Dynamic Velocimetry Enhancer* (DVE) module during the test phase for test-time optimization. This provides added flexibility to accommodate unseen situations and address potential inaccuracies arising from the limited training data context, which will be demonstrated in Sec. 4.4. In principle, our approach seeks a residual flow vector $\mathbf{R}$ such that $\mathbf{F} = \mathbf{F}_{\text{init}} + \mathbf{R}$, which can be optimized to rectify the inaccuracies. Formally, DVE is essentially an optimization process using the $L_{\text{recon}}$ objective function, with the formulation as follows:

$$\mathbf{R}^* = \underset{\mathbf{R} \in \mathbb{R}^{|\mathcal{Y}'| \times 3}}{\arg\min} \left( \frac{1}{|\mathcal{Y}'|} \sum_{\mathbf{y}_i' \in \mathcal{Y}'} p_i \min_{\mathbf{y}_j \in \mathcal{Y}} \|\mathbf{y}_i' + \mathbf{R}_i - \mathbf{y}_j\|_2^2 \right) \tag{4}$$

This test-time supervision (Fig. 2(g*)) is similar to $L_{\text{recon}} 1$ without the regularization $\lambda_{\text{conf}}$. Solved using an Adam optimizer, it only involves parameters from an $n_1 \times 3$ matrix. Concerning that existing test-time optimization modules [47] are slow, DVE is very efficient, as demonstrated later in Sec. 4.1.

**Selection of Self-supervised Losses During the Test Phase** We omit both $L_{\text{smooth}}$ and $L_{\text{div}}$ during the test stage. During the training phase, our objective is to embed prior knowledge about particle flow into the network. $L_{\text{smooth}}$ and $L_{\text{div}}$ serve not only to foster a comprehensive understanding of fluid behaviors but also function as regularizers, mitigating overfitting caused by the unconstrained reconstruction loss. However, in the test phase, our focus shifts to specific sparse particle sets.

In certain scenarios, such as when flows adhere to boundary conditions, these particles may not strictly adhere to the expected norms of ideal smoothness or zero-divergence typical in a flow field. Additionally, given that the initial flow estimate should be sufficiently accurate, regularizers become unnecessary. Our approach to customizing the loss functions in this manner aims to enhance the robustness of our model against the complex challenges encountered in real-world applications, **thereby improving data efficiency and cross-domain robustness.**

## 4    Experiments

We conduct comprehensive evaluations using different data domains on our proposed framework. First, we compare our method with SOTA fully supervised methods (Sec. 4.1). Next, we examine its performance under the constrained size of training data, reflecting real-world situations where domain-specific data is limited (Sec. 4.2). We then assess the framework's performance under different domains with increasing domain shift, highlighting its cross-domain robustness (Sec. 4.3). Additionally, we conduct comprehensive ablation studies on the components of our framework (Sec. 4.4) to validate their effects. Following the previous SOTA method GotFlow3D [52], our datasets include FluidFlow3D [52] and its six fluid cases, DeformationFlow [98] and AVIC [41]. **Due to page limit, experimental settings including implementation details, datasets, and evaluation metrics can be found in Appendix A.3.**

### 4.1    Comparison with state-of-the-art methods

Since DeepPTV[50] is not open-sourced, we enrich the comparison by including scene-flow methods [56; 70; 96; 92] as our baselines. We benchmark our method against established fully supervised models, such as FlowNet3D [57], FLOT [70], PointPWC-Net [96], PV-RAFT [92], and GotFlow3D [53], all utilizing the FluidFlow3D training set. All baseline models are evaluated using the default hyperparameters. As shown in Figure 4, our purely self-supervised approach outperforms all the fully supervised baselines. Additionally, we introduce *Ours (1%)*—our method trained on just 1% of the data—which still demonstrates comparable performance.

**Comparison Across Flow Cases:** We further analyze our framework's performance across six distinct flow cases from the FluidFlow3D dataset, with details available in Appendix A.3.1. For three representative cases—Uniform Flow, Turbulent Channel Flow, and Forced MHD Turbulence—we present detailed results in Fig. 4, while the remaining are documented in Appendix A.4.1. In the simple Uniform Flow case, our method shows slight improvement over the baseline. However, in more complex scenarios, such as Forced MHD Turbulence, our method significantly outperforms the baseline, reducing the EPE/NEPE metric by nearly half compared to the SOTA GotFlow3D.

**Test-time Efficiency:** In Figure 4, the $T_{\text{test}}$ column in the table illustrates the time consumption of each method during the test phase. Our method demonstrates time efficiency, incurring less inference time cost than even baseline supervised methods (without test-time optimization). Our method requires only a few epochs to converge (See Appendix A.5.3). Furthermore, our network's relatively small size (refer to $P_{\text{train}}$ comparison in Figure 4) facilitates a rapid forward pass. Time profiling is conducted on a single RTX 3090 Ti.

### 4.2    Training with Limited Data

Handling rare scenarios, such as unique geometric flow fields [88] or cytoplasmic flows in disease contexts, presents challenges in assembling large datasets. To address this, we explore an evaluation setup with limited training data (still without labels) by randomly sampling from the FluidFlow3D training dataset. We established three distinct training settings: a 100% sampling rate (13,621 samples), a 10% sampling rate (1,300 samples), and a 1% sampling rate (130 samples). Testing was conducted on the FluidFlow3D test data (see Appendix A.3.1). We compared our method with FLOT [70], PV-RAFT [92], and the current SOTA GotFlow3D [53]. We present the results of the major metric, EPE, with further details in Appendix A.4.2. Fig. 5(a) illustrates the robustness of our approach to reductions in training data size. Our metrics remain stable even with significant decreases in training samples, while other methods show substantial performance declines. This disparity becomes more pronounced in complex cases, as discussed below.

| Methods | $P_{train}$ | $T_{test}$ (s) | EPE ↓ | Acc Strict ↑ | Acc Relax ↑ | Outliers ↓ | NEPE ↓ |
|---|---|---|---|---|---|---|---|
| Flownet3D | 1.23 million | 0.478 | 0.0623 | 19.34% | 38.23% | 61.77% | 0.3452 |
| FLOT | **0.11 million** | 0.030 | 0.0587 | 24.99% | 45.59% | 54.41% | 0.3324 |
| PointPWC-Net | 7.72 million | 0.485 | 0.0172 | 46.75% | 71.61% | 28.39% | 0.0874 |
| PV-RAFT | 0.19 million | 1.079 | 0.0165 | 72.98% | 83.69% | 16.31% | 0.0898 |
| Gotflow3D | 1.44 million | 0.758 | 0.0049 | 93.15% | 96.38% | 3.62% | 0.0244 |
| Ours | 0.58 million | 0.218 | **0.0046** | **98.69%** | **98.77%** | **1.31%** | **0.0188** |
| Ours(1%) | 0.58 million | 0.218 | 0.0085 | 97.61% | 97.76% | 2.40% | 0.0317 |
| Ours(1%) w/o Div Loss | 0.58 million | 0.218 | 0.0090 | 97.45% | 97.60% | 2.56% | 0.0419 |
| Ours(1%) w/o DVE | 0.58 million | **0.019** | 0.0219 | 89.92% | 95.56% | 21.04% | 0.1087 |

| Cases | | | | | | Isometric View | End View |
|---|---|---|---|---|---|---|---|
| **Uniform Flow** | | | | | | | |
| Method | EPE↓ | Acc Strict↑ | Acc Relax↑ | Outliers↓ | NEPE↓ | | |
| FlowNet3D | 0.0375 | 25.41% | 79.23% | 20.77% | 0.075 | | |
| FLOT | 0.0206 | 68.25% | 96.45% | 3.55% | 0.043 | | |
| PointPWC-Net | 0.0214 | 64.31% | 97.32% | 2.68% | 0.044 | | |
| PV-RAFT | 0.0032 | **99.87%** | **99.99%** | **0.015%** | 0.007 | | |
| GotFlow3D | 0.0024 | 99.85% | 99.98% | 0.02% | 0.005 | | |
| Ours | **0.0018** | 99.42% | 99.44% | 0.58% | **0.004** | | |
| **Turbulent Channel Flow** | | | | | | | |
| Method | EPE↓ | Acc Strict↑ | Acc Relax↑ | Outliers↓ | NEPE↓ | | |
| FlowNet3D | 0.0367 | 24.89% | 70.78% | 29.22% | 0.086 | | |
| FLOT | 0.0354 | 26.89% | 72.72% | 27.25% | 0.083 | | |
| PointPWC-Net | 0.0127 | 87.51% | 98.41% | 1.59% | 0.032 | | |
| PV-RAFT | 0.0065 | 93.71% | 97.83% | 2.17% | 0.015 | | |
| GotFlow3D | 0.0024 | **99.13%** | **99.74%** | **0.26%** | 0.006 | | |
| Ours | **0.0019** | 99.38% | 99.41% | 0.62% | **0.005** | | |
| **Forced MHD Turbulence** | | | | | | | |
| Method | EPE↓ | Acc Strict↑ | Acc Relax↑ | Outliers↓ | NEPE↓ | | |
| FlowNet3D | 0.0940 | 0.06% | 0.45% | 99.55% | 0.851 | | |
| FLOT | 0.0984 | 0.04% | 0.28% | 99.72% | 0.876 | | |
| PointPWC-Net | 0.0171 | 8.46% | 33.53% | 66.47% | 0.165 | | |
| PV-RAFT | 0.0259 | 41.44% | 61.37% | 38.63% | 0.230 | | |
| GotFlow3D | 0.0060 | 83.01% | 91.81% | 8.19% | 0.053 | | |
| Ours | **0.0037** | **98.89%** | **98.95%** | **1.13%** | **0.031** | | |

Figure 4: (Top) **Benchmarking Against Fully Supervised Methods**. $P_{train}$ signifies the count of trainable parameters. $T_{test}$ stands for inference time for each sample. The best results are marked in bold. (Bottom) **Performance Across Flow Cases**. The best results are marked in bold, with the runners-up underlined. The subplots on the right visualize these three cases. The warmer color indicates a higher flow speed. All models are trained on full data, except Ours (1%).

**Performance Drop Across Flow Cases:** We further tested our method with limited training data on different flow cases from the FluidFlow3D test data mentioned above. The performance drop associated with limited data is shown in Fig. 5(b). As illustrated, complex flow cases such as Forced Isotropic Turbulence are more susceptible to limited data, while simpler flow cases like Uniform Flow and Turbulent Channel Flow maintain stable EPE as the training data decreases.

## 4.3 Analysis of Robustness Across Different Domains

Natural fluids exhibit a range of behaviors, including convection and laminar flow. Gathering data under all possible conditions presents significant challenges. To address this, we examine the cross-domain knowledge transfer capability of our proposed method. We explore a gradual increase in domain shift: initially, we investigate fluid case domain shifts (Sec. 4.3.1), where we train on five

| Method\Flow Cases | Beltrami | Channel | Isotropic | Mhd | Transition | Uniform |
|---|---|---|---|---|---|---|
| Ours | **0.01760** | **0.00250** | **0.01840** | **0.00510** | **0.00200** | **0.00200** |
| GotFlow3D | 0.03263 | 0.01899 | 0.05570 | 0.03422 | 0.01076 | 0.02439 |

(a)

| Method\Train Size | 100% | 10% | 1% |
|---|---|---|---|
| FLOT | 0.05870 | 0.08050 | 0.12954 |
| PV-RAFT | 0.01650 | 0.06298 | 0.18097 |
| GotFlow3D | 0.00487 | 0.02032 | 0.02773 |
| Ours | **0.00460** | **0.00640** | **0.00850** |

(b)

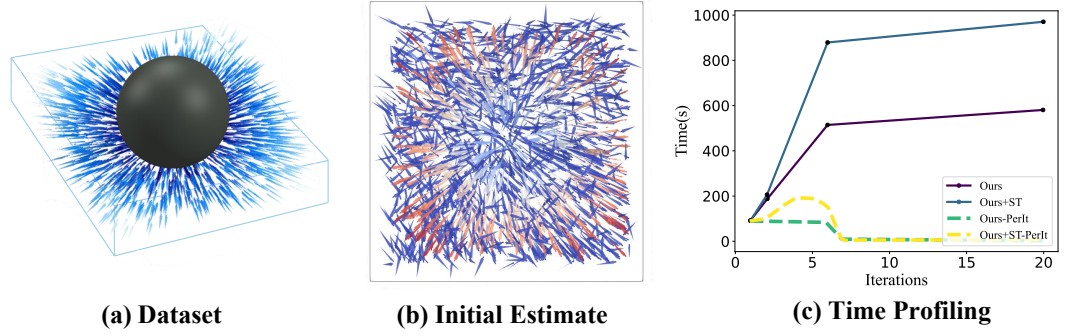

(c)

Figure 5: (a) Leave-one-out domain EPE Comparison: "Flow Cases" stands for the flow case we test on with the model trained on the rest five cases. (b) Comparison of EPE with Limited Training Data. (c) Performance Drop related to Limited Training Data. The Y-axis shows the major matric EPE, and the X-axis indicates the percentage of the training dataset utilized.

**(a) Dataset**      **(b) Initial Estimate**      **(c) Time Profiling**

Figure 6: (a) DeformationFlow data. (b) Initial estimation by our method. (c) Time-consumption comparison between *SerialTrack* and *Ours+ST*. "PerIt" denotes time per PTV iteration.

certain fluid cases and test on the leave-one-out domain within the same dataset. Next, we examine the Sim2Real domain shift (Sec. 4.3.2), where training occurs on a synthetic fluid dataset and testing on real-world fluid data. Lastly, we assess a more extensive Sim2Real domain shift (Sec. 4.3.3) by training on synthetic physical fluid data and testing on biological datasets.

**4.3.1 Testing within the Same Synthetic Fluid Dataset:** In this section, we employ the complete FluidFlow3D training set in a six-fold cross-validation setup, training on five sub-cases and testing on the remaining one. We benchmark our method solely against the state-of-the-art fluid motion learning method, GotFlow3D [53], as other baselines are not fluid-specific. The EPE metric results, shown in Fig.5(c) (with additional results in Appendix A.4.3), indicate that our method outperforms GotFlow3D in various scenarios, especially in complex conditions like MHD and isotropic turbulence. Moreover, our method shows consistent performance, highlighting its robustness in unfamiliar scenarios.

*Sim2Real Experimental Setting:* Synthetic data with ground-truth labels often serves as a benchmark for method evaluation. However, the domain gap between synthetic and real data can negatively impact performance, underscoring the importance of validation on real-world datasets. In this and the following section, we validate our method using two real-world datasets, DeformationFlow [98] and Aortic Valve Interstitial Cell (AVIC) [41] (details in Appendix A.3.1), to demonstrate its practical application potential. Two challenges in real-world evaluation are: 1) the lack of ground-truth labels in real-world data, and 2) the requirement for a complete PTV method for particle tracking application. Therefore, we use the following setting: our method, trained on FluidFlow3D, is integrated into PTV algorithms (specified below) to provide initial motion estimates. Due to the absence of ground truth, we primarily demonstrate the generalizability of our method across various domains through qualitative results. Quantitatively, we emphasize efficiency in the physical domain, where multiple frames are involved. By contrast, in the biological domain, we focus on validating the plausibility of our estimates, especially given the significant cell deformation, where efficiency is less critical.

Table 1: Comparison on AVIC data. C2E, C2N, E2N stands for 3 settings: Cyto-D treatment to Endo-1 treatment, Cyto-D treatment to Normal and Endo-1 treatment to Normal. MNDS stands for the mean neighbor distance score.

| Method | Tracked Matches ↑ | | | MNDS ↓ | | |
|--------|------|------|------|-------|-------|-------|
|        | C2E  | C2N  | E2N  | C2E   | C2N   | E2N   |
| Fm-track | 8040 | 7744 | 8097 | 0.358 | 0.283 | 0.399 |
| Ours+Fm  | 8048 | 7750 | 8097 | 0.357 | 0.246 | 0.359 |

**4.3.2 Testing from Synthetic to Real-World Fluid Data:** We employ SerialTrack (ST) [98] as the PTV framework and designate our integrated version as *Ours+ST*. Quantitative results highlight the advantages of our method: it identifies 22,882 matches, exceeding the 22,001 particles tracked by vanilla SerialTrack, and significantly reduces tracking time by providing accurate initial estimates that expedite match finding. Results in Fig. 6 showcase the initial estimations and time efficiency of *Ours+ST*. These outcomes affirm that our method delivers sufficiently precise estimations to improve PTV, demonstrating strong simulation-to-reality (Sim2Real) capabilities.

**4.3.3 Testing from Synthetic Physical Fluids to Biological Data:** This study analyzes a dataset of AVIC images embedded in a PEG hydrogel, using microspheres to track hydrogel movements. AVICs were subjected to three conditions: regular, Cyto-D exposure, and Endo 1 treatment. Cell deformation, challenging to observe directly, was quantified by tracking nearby particles with *Fm-track* [41], using our framework for initialization. Results are presented in Tab.1, and a visualization of the cell deformation we estimate is in AppendixA.4.4. We evaluate particle movement using the neighbor distance score (See Appendix A.3.2), with higher scores indicating less accurate estimations. Our results in Tab. 1 show *Ours+Fm* slightly outperforming *Fm-track*. Notably, our method, trained exclusively on the FluidFlow3D Dataset, demonstrates strong adaptability across domains and provides insights into biological fluid dynamics.

### 4.4 Ablation study on different modules.

In addition to the aforementioned assessments, an ablation study regarding different proposed modules including different feature extractors, Zero-divergence Loss, and DVE is performed to demonstrate their effectiveness. We show our method w/ and w/o Div Loss (Zero-Divergence Loss) and DVE module in Fig. 4. The full results are illustrated in Appendix A.5.

## 5 Conclusion

In this paper, we introduce a test-time self-supervised framework for learning 3D fluid motion from dual-frame unstructured particle sets. We address the challenge of improving data efficiency and ensuring cross-domain robustness, which are crucial for practical applications. We demonstrate the viability of our approach through two real-world studies and suggest that our findings could inform further research into extensive real-world applications, the exploration of constraints specific to particular scenarios, and the development of novel model architectures for enhanced adaptability.

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

# A Appendix

## A.1 Methods in Detail

### A.1.1 Feature extractor

We draw significant inspiration from GotFlow3D's feature extractor [52] to obtain a robust representation of particle features. We briefly describe the structure of the feature extractor here.

The feature extractor is based on a graph neural network. Initially, a static graph $G_{\text{static}} = (V_s, E_s)$ is established in the 3D spatial space for the input point cloud. $V_s$ contains the given input points, while $E_s$ consists of connections between the k-nearest neighbors of each point. This process yields the original point cloud feature $f_i$ for point $i$ as a hexadecimal set, including the three-dimensional coordinates, the radial distance, the azimuthal angle, and the polar angle. After processing through several GeoSetConv layers [72], we obtain a high-dimensional geometric local feature:

$$F_i^n = \text{MaxPooling}_{j \in N_k(i)}\{\Phi^n(f_i, F_j^{n-1} - F_i^{n-1})\},$$

where $F_i^n$ denotes the feature extracted from the static graph at the $n$-th layer for point $i$, and $N_k(i)$ represents its k-nearest neighbors in the static graph. $\Phi^n$ stands for the $n$-th GeoSetConv layer. $F_i^0$ is initialized by the 3D coordinates of the input point cloud.

Subsequently, the dynamic graph is generated. It takes these high-dimensional features $F_i$ as inputs and, following the same steps as the construction of the static graph, seeks the nearest neighbors based on these features to establish connections, thus obtaining the local structure on the feature manifold. However, the distinction lies in the fact that the dynamic graph possesses a greater receptive range since it accepts high-dimensional features. Moreover, it is reconstructed during every training session, whereas a static graph is constructed only once.

The dynamic feature is obtained in the same manner as $F_i$, with the sole difference being the use of EdgeConv layers [91] instead of GeoSetConv layers. The final feature is obtained by concatenating all hierarchical features from different layers of both static and dynamic graphs.

## A.2 Optimal Transport for Soft Correspondence

Let $\Phi_{\mathcal{X}}$ and $\Phi_{\mathcal{Y}}$ represent the features extracted from two input particle sets, $\mathcal{X}$ and $\mathcal{Y}$, respectively. We initiate by computing the point-to-point similarity matrix, given by:

$$\mathbf{S}_{i,j} = \frac{\Phi_{\mathbf{x}_i} \cdot \Phi_{\mathbf{y}_j}^T}{\|\Phi_{\mathbf{x}_i}\|_2 \|\Phi_{\mathbf{y}_j}\|_2} \tag{5}$$

In this context, $\Phi_{\mathbf{x}_i}$ pertains to the feature of the $i$-th point in $\Phi_X$, and analogously for $\Phi_{\mathbf{y}}^j$.

Inspired by pioneering research, we formulate the correspondence linking problem through the framework of optimal transport [39]. Assigning a mass of $\frac{1}{|\mathcal{X}|}$ to each source point $\mathbf{x}_i$, we consider its transport to the target point $\mathbf{y}_j$ with the cost matrix defined by $\mathbf{C}_{i,j} = 1 - \mathbf{S}_{i,j}$. In this context, a higher cost indicates a lower similarity between two points within the feature space. Our objective is to identify the optimal transport plan $\mathbf{T}^*$ that satisfies,

$$\mathbf{T}^* = \arg\min_{\mathbf{T} \in \mathbb{R}_+^n} \left[ \sum_{i,j} \mathbf{T}_{i,j} \mathbf{C}_{i,j} + \epsilon \sum_{i,j} \mathbf{T}_{i,j} (\log \mathbf{T}_{i,j} - 1) \right.$$
$$\left. + \lambda \left( \text{KL}(\mathbf{T1}, \frac{1}{|\mathcal{X}|}\mathbf{1}) + \text{KL}(\mathbf{T}^T\mathbf{1}, \frac{1}{|\mathcal{X}|}\mathbf{1}) \right) \right], \tag{6}$$

where KL denotes the Kullback-Leibler (KL) divergence, $\mathbf{1} \in \mathbb{R}^{|\mathcal{X}| \times 1}$ is a vector filled with ones, and the terms involving $\epsilon$ and $\lambda$ are regularizing terms with coefficients controlling their respective strengths.

The optimal transport plan $\mathbf{T}^*$ yields the soft correspondence weight of point $x_i$ to point $y_j$ as:

$$\mathbf{W}_{i,j} = \frac{e^{\mathbf{T}_{i,j}^*}}{\sum_{k \in \mathcal{M}_{\mathcal{Y}}(\mathbf{x}_i)} e^{\mathbf{T}_{i,k}^*}}, \tag{7}$$

where $\mathcal{M}_{\mathcal{Y}}(\mathbf{x}_i)$ represents the set of $L$ points from $\mathcal{Y}$ corresponding to the top $L$ values of $\mathbf{T}_{i,j}^*$, and $L$ is a hyper-parameter that can be chosen for specific particle tracking case. Consequently, we can estimate the new location of a given $x_i$, the estimated position is given by,

$$\mathbf{y}_i^* = \sum_{\mathbf{y}_j \in \mathcal{Y}} W_{i,j} \mathbf{y}_j. \tag{8}$$

And the confidence score $p_i$, which quantifies the reliability of the estimated position $\mathbf{y}_i^*$ for each point $\mathbf{x}_i$ is given by,

$$p_i = \max \left( \sum_{k \in \mathcal{M}_{\mathcal{Y}}(\mathbf{x}_i)} \mathbf{W}_{i,k} \mathbf{S}_{i,k}, 0 \right). \tag{9}$$

Finally, the flow estimate $f_i$ is determined as,

$$\mathbf{f}_i = \mathbf{y}_i^* - \mathbf{x}_i. \tag{10}$$

## A.3  Experimental Setup

### A.3.1  Datasets

**FluidFlow3D Dataset**   The FluidFlow3D [53] is a large synthetic dataset designed for the study of 3D fluid flow. Specifically, it offers enough data for training and serves as a benchmark to evaluate the flow estimation capabilities of supervised 3D fluid flow motion learning techniques. This dataset utilizes physically accurate simulated flow structures sourced from public database [49], ensuring that the provided ground truth flows adhere to computational fluid dynamics principles. It encompasses six typical categories of flow cases, namely, uniform flow, isotropic turbulent flow, magneto-hydrodynamic (MHD) turbulence, fully developed turbulent channel flow, transitional boundary layer flow, and the Beltrami flow [17]. We utilize this synthetic dataset to evaluate our method, comparing it with baselines (see Sec. 4.1), training with restricted data capacity (see Sec. 4.2), or testing cross-domain transferability by training on and evaluating different turbulence types (see Sec. 4.3). It should be noted that, to our knowledge, this dataset is the only large-scale benchmark specifically designed for dual-frame fluid motion learning. Other fluid datasets, such as *CylinderFlow*[34], focus on fluid trajectories across multiple frames, requiring the integration of the full PTV process. This approach deviates from the main problem we aim to address.

**DeformationFlow**   The DeformationFlow dataset [98] showcases real-world physical dynamics by recording the indentation created when a stainless steel sphere is placed on a soft polyacrylamide hydrogel due to gravitational forces. Fluorescent fluid particles within the gel were scanned both pre and post-indentation, yielding 3D volumetric images. We employ this dataset because it diverges significantly from the synthetic training set. **Notably, the zero-divergence constraint is no longer applicable**, making it an ideal candidate to evaluate the cross-domain generalization capability of our proposed test-time self-supervised framework.

**Aortic Valve Interstitial Cell (AVIC) Dataset**   The AVIC dataset [41] represents real-world biological dynamics revealed through PTV analysis. In the dataset, AVICs, extracted from dissected porcine heart leaflets, were suspended in a peptide-modified PEG hydrogel with fluorescent microbeads and subsequently incubated for 72 hours. We employ this dataset to demonstrate the strong cross-domain versatility of our proposed framework with the data from the biological realm.

### A.3.2  Evaluation Metrics

For a thorough evaluation of our framework's performance and its comparison with previous works, we utilize five prominent metrics commonly applied in particle tracking velocimetry evaluations: **EPE**, **NEPE**, **Acc Strict**, **Acc Relax**, and **Outliers**.

To introduce the metrics, we first define point error $e_i$ and the relative point error $e_i^{\text{rel}}$,

$$e_i = \|\mathbf{f}_i^* - \mathbf{f}_i^{\text{gt}}\|_2, \quad e_i^{\text{rel}} = \frac{e_i}{\|\mathbf{f}_i^{\text{gt}}\|_2}, \tag{11}$$

where $e_i$ represents the Euclidean distance (L2 norm) between the predicted flow $\mathbf{f}_i^*$, and the ground-truth flow $\mathbf{f}_i^{\text{gt}}$, for a specific point $\mathbf{x}_i$. The relative error $e_i^{\text{rel}}$ provides a normalized measure, indicating the magnitude of the point error in relation to the magnitude of the ground-truth flow at point $\mathbf{x}_i$.

Then, the metrics used for evaluating the quality of predicted flows are defined as follows. The **EPE** (End Point Error), calculated as $\text{EPE} = \frac{1}{|\mathcal{X}|} \sum_{\mathbf{x}_i \in \mathcal{X}} e_i$, representing the average point errors, and the **NEPE** (Normalized End Point Error), given by $\text{NEPE} = \frac{1}{|\mathcal{X}|} \sum_{\mathbf{x}_i \in \mathcal{X}} e_i^{\text{rel}}$, reflecting the normalized average of point errors. Furthermore, the **Acc Strict** metric denotes the percentage of points with $e_i < 0.05[m]$ or $e_i^{\text{rel}} < 5\%$, while **Acc Relax** captures the points having $e_i < 0.10[m]$ or $e_i^{\text{rel}} < 10\%$. Lastly, **Outliers** indicates points where $e_i > 0.30[m]$ or $e_i^{\text{rel}} > 10\%$, signifying notable deviations between predictions and ground truth, a potential indicator of the model's challenges in generalizing across diverse data.

Furthermore, when performing experiments on real datasets, the absence of ground truth in actual data prompts us to incorporate additional metrics for evaluating our methodology. For the *SerialTrack* assessment, our evaluative metrics include **Matches**, **Tracking ratio**, **UpdateNorm**, and **Time**. We use $n_{\text{match}}$ to represent the matches identified between the source and target point clouds, and $n_{\text{no\_missing}}$ signifies the count of source points that are matched in the target point cloud. Consequently, **Matches** is equivalent to $n_{\text{match}}$, and **Tracking ratio** is expressed as the fraction $\frac{n_{\text{match}}}{n_{\text{no\_missing}}}$. The **UpdateNorm** metric captures the change in PTV parameters for each iteration, while **Time** measures the duration taken for a single iteration. For the *Fm-track* assessment, our primary metric is the **neighbor distance score** (NDS), delineated for a point $\mathbf{x}_i$ as,

$$\text{NDS}_i = \frac{1}{N} \sum_{\mathbf{x}_j \in N(\mathbf{x}_i)} \|\mathbf{f}_i - \mathbf{f}_j\|_2^2, \tag{12}$$

where $N(\mathbf{x}_i)$ is the K-nearest-neighbor set of point $\mathbf{x}_i$, $\mathbf{f}_i$ denotes the flow vector at point $\mathbf{x}_i$.

### A.3.3 Implementation Details

In our method, there are some pre-defined hyperparameters. For the sake of experimental reproducibility, we list them below.

**MODEL STRUCTURE** For the feature extractor, the K-value of our K-nearest-neighbor is chosen to be 32, the embedding-dim to be 128, and the dropout rate to be 0.5. The grid size for splatting is $10 \times 10 \times 10$.

**LOSS TERM** The selected number of neighboring points for the reconstruction loss $L_{\text{recon}}$ is 32. Likewise, the number of neighboring points for smooth flow loss $L_{\text{smooth}}$ is 32, and for zero-divergence loss $L_{\text{div}}$ is 2. $\lambda_{\text{conf}}$ is 0.1, $\lambda_{\text{smooth}}$ is 10, and $\lambda_{\text{div}}$ is 0.1.

During the training phase, we utilize a mini-batch training process with a batch size of 4. To achieve convergence, we train the full-data model and 10%-data model for 100 epochs, and 1%-data model for 300 epochs. A default learning rate of 0.001 is set and the training is run on a single RTX 4070TI.

During the test phase, DVE runs for 150 steps with an update rate of 0.01.

### A.4 Extended Results

#### A.4.1 Full Comparison with the state-of-the-art methods on different flow cases of FluidFlow3D dataset

We present a comprehensive comparison with state-of-the-art methods across all flow cases in Table 2. Notably, our model, trained on only 130 samples, outperforms the current state-of-the-art, GotFlow3D, and is marked in italics. In particularly complex scenarios such as Forced MHD turbulence and Forced isotropic turbulence, our method significantly outperforms GotFlow3D, even when trained with only 1% of the training set samples. In more common scenarios, our model, trained with the full dataset, surpasses GotFlow3D in most metrics, while the model trained with limited samples still demonstrates commendable performance.

#### A.4.2 Full Results of the Training with Limited Data Experiment

We present the full results of our limited-data training experiment here in Tab. 3.

Table 2: This table illustrates the performance of our method relative to baseline methods, with the size of the training set indicated in parentheses after each technique—1300 corresponds to 10% of the full dataset, and 130 corresponds to 1%. The leading results are emphasized in bold, while the second-best ones are underlined.

| Method | Beltrami Flow | | | | | Method | Turbulent Channel Flow | | | | |
|---|---|---|---|---|---|---|---|---|---|---|---|
| | EPE | Acc Strict | Acc Relax | Outliers | NEPE | | EPE | Acc Strict | Acc Relax | Outliers | NEPE |
| FlowNet3D(all) | 0.04435 | 2.30% | 13.73% | 86.27% | 0.273 | FlowNet3D(all) | 0.0367 | 24.89% | 70.78% | 29.22% | 0.086 |
| FLOT(all) | 0.02417 | 13.11% | 42.22% | 57.78% | 0.169 | FLOT(all) | 0.0354 | 26.89% | 72.72% | 27.25% | 0.083 |
| PointPWC-Net(all) | 0.01616 | 24.36% | 62.19% | 37.81% | 0.109 | PointPWC-Net(all) | 0.0127 | 87.51% | 98.41% | 1.59% | 0.032 |
| PV-RAFT(all) | 0.00899 | 69.69% | 91.00% | 9.00% | 0.047 | PV-RAFT(all) | 0.0065 | 93.71% | 97.83% | 2.17% | 0.015 |
| GotFlow3D(all) | **0.00291** | 95.03% | 98.44% | 1.56% | **0.018** | GotFlow3D(all) | 0.00241 | 99.13% | **99.74%** | **0.26%** | 0.006 |
| Ours(all) | 0.0039 | **98.95%** | **98.99%** | **1.05%** | 0.0191 | Ours(all) | **0.0019** | **99.38%** | 99.41% | 0.62% | **0.0051** |
| Ours(1300) | 0.0064 | 98.39% | 98.45% | 1.62% | 0.0243 | Ours(1300) | 0.0024 | 99.31% | 99.35% | 0.69% | 0.0063 |
| Ours(130) | 0.0101 | 97.58% | 97.66% | 2.44% | 0.0319 | Ours(130) | 0.0025 | 99.25% | 99.31% | 0.75% | 0.0065 |

| Method | Forced Isotropic Turbulence | | | | | Method | Forced MHD Turbulence | | | | |
|---|---|---|---|---|---|---|---|---|---|---|---|
| | EPE | Acc Strict | Acc Relax | Outliers | NEPE | | EPE | Acc Strict | Acc Relax | Outliers | NEPE |
| FlowNet3D(all) | 0.12465 | 0.20% | 1.52% | 98.48% | 0.558 | FlowNet3D(all) | 0.0940 | 0.06% | 0.45% | 99.55% | 0.851 |
| FLOT(all) | 0.13090 | 0.14% | 1.15% | 98.85% | 0.587 | FLOT(all) | 0.0984 | 0.04% | 0.28% | 99.72% | 0.876 |
| PointPWC-Net(all) | 0.02719 | 18.47% | 57.06% | 42.94% | 0.116 | PointPWC-Net(all) | 0.0171 | 8.46% | 33.53% | 66.47% | 0.165 |
| PV-RAFT(all) | 0.04190 | 51.37% | 62.82% | 37.18% | 0.176 | PV-RAFT(all) | 0.0259 | 41.44% | 61.37% | 38.63% | 0.230 |
| GotFlow3D(all) | **0.01153** | 86.62% | 90.74% | 9.26% | 0.045 | GotFlow3D(all) | 0.00596 | 83.01% | 91.81% | 8.19% | 0.053 |
| Ours(all) | 0.0117 | **96.94%** | **97.04%** | **3.07%** | **0.0378** | Ours(all) | **0.0037** | **98.89%** | **98.95%** | **1.13%** | **0.0306** |
| Ours(1300) | 0.0156 | 96.11% | 96.26% | 3.92% | 0.0515 | Ours(1300) | 0.0051 | 98.51% | 98.59% | 1.53% | 0.0412 |
| Ours(130) | 0.0193 | 95.01% | 95.22% | 5.02% | 0.0645 | Ours(130) | 0.0066 | 97.98% | 98.12% | 2.06% | 0.0505 |

| Method | Transitional Boundary Flow | | | | | Method | Uniform Flow | | | | |
|---|---|---|---|---|---|---|---|---|---|---|---|
| | EPE | Acc Strict | Acc Relax | Outliers | NEPE | | EPE | Acc Strict | Acc Relax | Outliers | NEPE |
| FlowNet3D(all) | 0.02003 | 67.23% | 91.01% | 8.99% | 0.048 | FlowNet3D(all) | 0.0375 | 25.41% | 79.23% | 20.77% | 0.075 |
| FLOT(all) | 0.01715 | 70.37% | 94.63% | 5.37% | 0.043 | FLOT(all) | 0.0206 | 68.25% | 96.45% | 3.55% | 0.043 |
| PointPWC-Net(all) | 0.01163 | 89.13% | 98.29% | 1.71% | 0.030 | PointPWC-Net(all) | 0.0214 | 64.31% | 97.32% | 2.68% | 0.044 |
| PV-RAFT(all) | 0.00324 | 99.65% | 99.94% | 0.063% | 0.008 | PV-RAFT(all) | 0.0032 | **99.87%** | 99.99% | 0.015% | 0.007 |
| GotFlow3D(all) | **0.00222** | **99.87%** | **99.97%** | **0.03%** | **0.005** | GotFlow3D(all) | 0.00244 | 99.85% | 99.98% | 0.02% | 0.005 |
| Ours(all) | 0.0028 | 99.02% | 99.18% | 0.96% | 0.0067 | Ours(all) | **0.0018** | 99.42% | 99.44% | 0.58% | **0.004** |
| Ours(1300) | 0.004 | 98.60% | 98.81% | 1.37% | 0.0093 | Ours(1300) | 0.002 | 99.40% | 99.42% | 0.60% | 0.0043 |
| Ours(130) | 0.0066 | 97.55% | 97.87% | 2.42% | 0.0149 | Ours(130) | 0.0019 | 99.41% | 99.43% | 0.59% | 0.0041 |

### A.4.3 Full Results of the Cross-Domain Robustness Experiment within the Same Synthetic Fluid Dataset

We present the full results of our cross-domain robustness experiment here in Fig. 7. Our technique consistently outperforms the state-of-the-art method GotFlow3D across different fluid scenarios in terms of accuracy. Furthermore, our method demonstrates a consistent performance across these scenarios, unlike the pronounced variability observed with GotFlow3D. This suggests that, after training on certain fluid cases, our model can proficiently handle scenarios it hasn't directly observed. This adaptability can be attributed to the DVE module, which allows for on-the-fly adjustments to unseen examples during testing.

### A.4.4 Full results of the AVIC experiment

**Experimental Settings:** In this work, we analyzed a dataset containing images of porcine aortic valve interstitial cells (AVICs) embedded in a polyethylene glycol (PEG) hydrogel. We employed 0.5 [$\mu m$] microspheres as tracking particles to monitor the movements within the PEG hydrogel. For detailed information on AVIC encapsulation within PEG hydrogels, readers are referred to a previous study [33]. We subjected the AVICs to three distinct conditions: initially, under regular conditions; secondly, after exposure to Cytochalasin-D, a compound that disrupts actin polymerization and cellular contraction; and thirdly, following treatment with Endo 1. Direct observation of cell deformation being challenging, we tracked the motion of particles adjacent to the cells within the gel under different treatments as a proxy measure. We utilized Fm-track[41], a specialized particle tracking velocimetry (PTV) method for cell tracking, providing it with an initialization value.

Table 3: Comparative performance of different methods under varying training data sizes. The table lists End-Point Error (EPE), Accuracy Strict, Accuracy Relaxed, and Outliers percentages for each method (FLOT, PV-RAFT, Gotflow3D, and Ours) at 100%, 10%, and 1% training data utilization. This data highlights the resilience of our method against reductions in training size, maintaining high accuracy and low outlier rates across all sampling levels.

| Method | Train Size | EPE | Acc Strict | Acc Relax | Outliers |
|---|---|---|---|---|---|
| FLOT | 100% | 0.05870 | 24.99% | 45.59% | 54.41% |
| | 10% | 0.08050 | 43.65% | 74.10% | 73.13% |
| | 1% | 0.12954 | 11.55% | 45.13% | 92.13% |
| PV-RAFT | 100% | 0.01650 | 72.98% | 83.69% | 16.31% |
| | 10% | 0.06298 | 45.83% | 82.87% | 68.03% |
| | 1% | 0.18097 | 7.11% | 29.82% | 95.62% |
| Gotflow3D | 100% | 0.00487 | 93.15% | 96.38% | 3.62% |
| | 10% | 0.02032 | 46.63% | 66.31% | 33.69% |
| | 1% | 0.02773 | 33.66% | 57.08% | 42.92% |
| Ours | 100% | 0.00460 | 98.69% | 98.77% | 1.31% |
| | 10% | 0.00640 | 98.27% | 98.37% | 1.74% |
| | 1% | **0.00850** | **97.61%** | **97.76%** | **2.40%** |

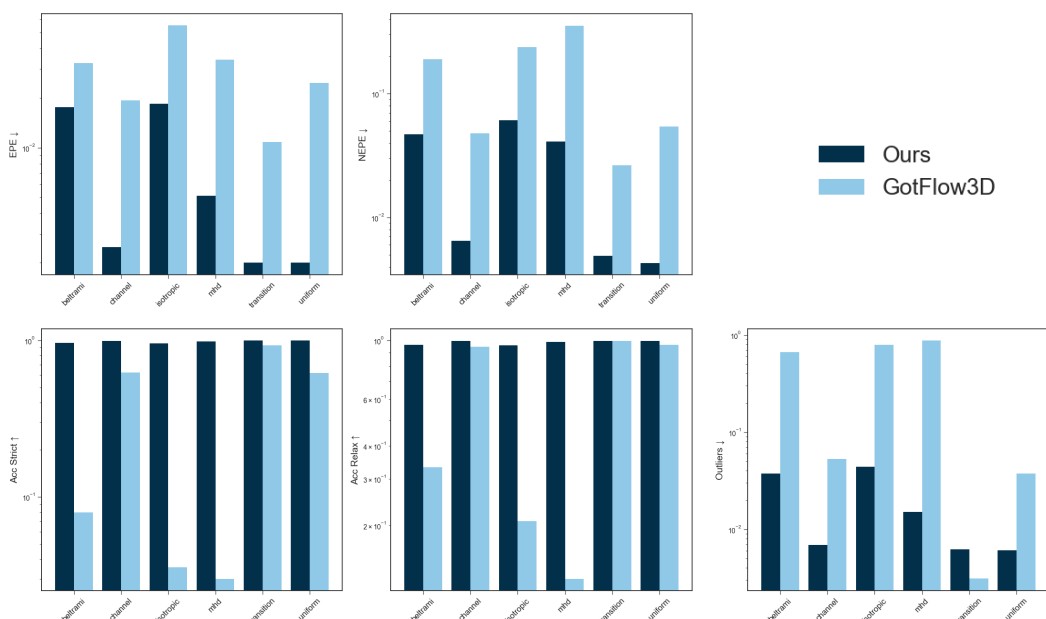

Figure 7: **Cross-domain Robustness Analysis.** Our method is compared with the state-of-the-art method GotFlow3D. Both techniques are trained on five flow cases and evaluated on the rest different case, which is indicated on the X-axis. The Y-axis showcases the metric of evaluation.

To illustrate the estimated cellular deformation before and after treatment with Cytochalasin-D, we provide a visualization in Figure 8(a). This figure successfully captures the distinctions between untreated cells and those treated with Cytochalasin-D. The left-hand side of Figure 8(a) displays the gel particle flow field, where the color indicates the angle between the cell surface's normal and the

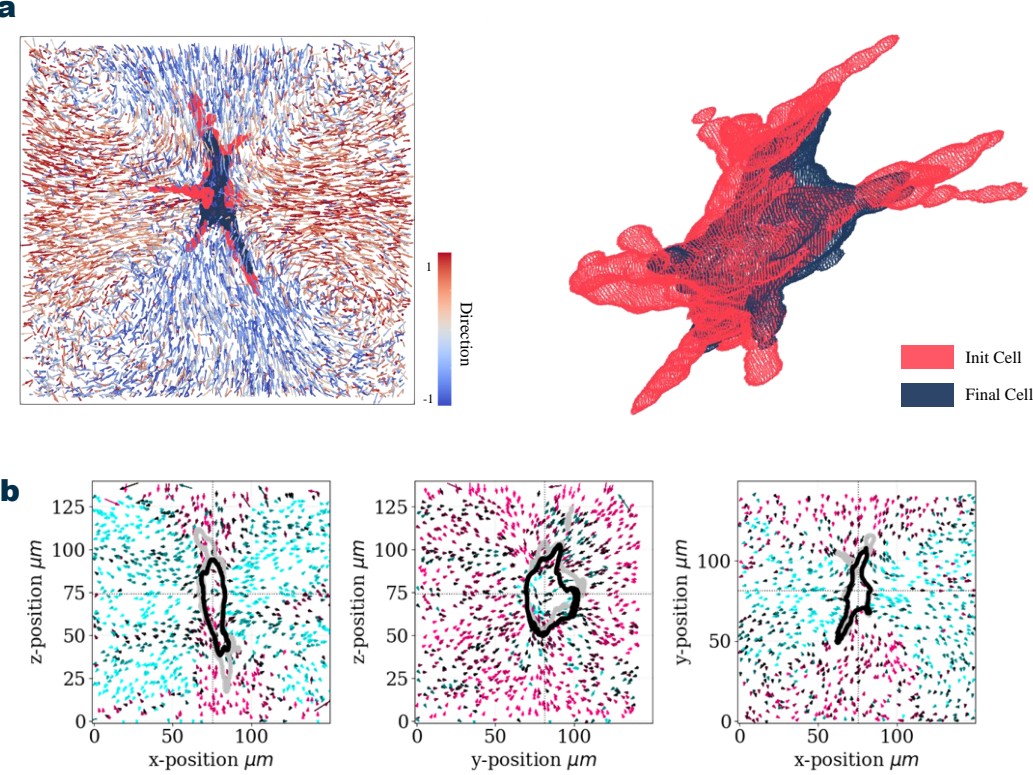

Figure 8: **Visualization of Our Results on AVIC.** (a) Enhanced Fm-track using our method illustrates cellular deformation pre and post-treatment with Cytochalasin-D. (b) Display of flow field profiles across the x, y, and z axes.

flow vector. In the central region of this flow field, which is further magnified on the right-hand side of the subplot, a directional flow and specific vortex patterns are evident, attributes linked to cell contraction. Further analysis is provided in Figure 8(b), which elaborates on the flow field profiles across the $x$, $y$, and $z$ axes, shedding light on the degree of cellular deformation in each dimension and aiding in the interpretation of the observed biological phenomenon.

### A.4.5 Subset Selection Robustness

Different data subset selection may influence our results on limited data. We control experiment randomness with seeds, conducting multiple runs to show our model performance on various subset data. See results in Table. 4.

## A.5 Ablation study on different modules

### A.5.1 Feature extractor

We compared two feature extractors: our graph-based feature extractor and Pointnet++ [72], a generic point cloud feature extractor, to assess their efficiency in extracting features from fluid particles. Both extractors have an embedding dimension of 128 and use a K-value of 32 for the K-nearest-neighbor algorithm. The results in Tab. 5 show that our feature extractor consistently outperforms Pointnet++ across all metrics, indicating that Pointnet++ has limitations in fully capturing the features of fluid particles.

Table 4: Various runs with different seeds.

| Seed | Train Size | EPE | Acc Strict | Acc Relax | Outliers |
|------|-----------|---------|-----------|-----------|----------|
| 42   | 100%      | 0.00460 | 98.69%    | 98.77%    | 1.31%    |
|      | 10%       | 0.00640 | 98.27%    | 98.37%    | 1.74%    |
|      | 1%        | 0.00850 | 97.61%    | 97.76%    | 2.40%    |
| 0    | 100%      | 0.00470 | 98.72%    | 98.77%    | 1.28%    |
|      | 10%       | 0.00530 | 98.56%    | 98.63%    | 1.45%    |
|      | 1%        | 0.00810 | 97.92%    | 98.01%    | 2.09%    |
| 1    | 100%      | 0.00490 | 98.66%    | 98.72%    | 1.35%    |
|      | 10%       | 0.00530 | 98.56%    | 98.63%    | 1.45%    |
|      | 1%        | 0.00760 | 98.05%    | 98.14%    | 1.97%    |
| 2    | 100%      | 0.00430 | 98.79%    | 98.84%    | 1.22%    |
|      | 10%       | 0.00600 | 98.36%    | 98.43%    | 1.66%    |
|      | 1%        | 0.00780 | 97.94%    | 98.03%    | 2.08%    |
| 3    | 100%      | 0.00450 | 98.76%    | 98.81%    | 1.25%    |
|      | 10%       | 0.00550 | 98.53%    | 98.59%    | 1.49%    |
|      | 1%        | 0.00720 | 98.07%    | 98.16%    | 1.95%    |
| 4    | 100%      | 0.00440 | 98.77%    | 98.82%    | 1.24%    |
|      | 10%       | 0.00580 | 98.44%    | 98.51%    | 1.57%    |
|      | 1%        | 0.00790 | 97.95%    | 98.04%    | 2.07%    |

Table 5: Comparison between our feature extractor with Pointnet++. Both models are trained on the full Fluidflow3D dataset.

| Feature Extractor | EPE | Acc Strict | Acc Relax | Outliers |
|-------------------|--------|-----------|-----------|----------|
| Ours              | 0.0046 | 98.69%    | 98.77%    | 1.31%    |
| Pointnet++        | 0.0583 | 57.57%    | 81.98%    | 59.65%   |

### A.5.2 Zero-Divergence Loss

When investigating the intrinsic properties of the velocity field, it is understood that incompressible fluids, by definition, exhibit zero divergence. Consequently, we have introduced a regulation specifically targeting zero-divergence in the estimated velocity field. However, this regulation is not entirely accurate, as many fluids can be compressed. Therefore, this section scrutinizes the effectiveness of the zero-divergence regulation. We tested it using the FluidFlow3D test data(See Table. 6) and its six fluid cases(See Table 7).

As shown in Table 6, the zero-divergence regulation appears ineffective for models trained on full samples when tested on the FluidFlow3D-NORM dataset. However, this regulation becomes more effective for models trained on fewer samples. Furthermore, the lower the sample count, the greater the benefit gained from zero-divergence regulation. Our interpretation is that the zero-divergence regulation provides additional prior information, which can be learned with a sufficiently large sample size but may be challenging to access when the sample size is limited. Thus, our zero-divergence regulation plays a compensatory role that improves data efficiency.

We conducted further tests to determine the suitability of the zero-divergence assumption for various fluid cases. Our findings, illustrated in Figure 7, revealed that our regulation performs excellently in fluid cases that exhibit zero divergence, such as Beltrami Flow, Turbulent Channel Flow, Transitional Boundary Flow, and Uniform Flow. Specifically, our model trained with 130 samples (with zero-divergence regulation) outperformed an unregulated model trained with 1300 samples across all metrics in Uniform Flow. In more complex fluid cases, such as Forced MHD Turbulence, where the zero-divergence law does not hold, employing zero-divergence may capture incorrect fluid features if the model is trained with insufficient data. However, with a full training dataset, it still enhances performance.

Table 6: Comparison on FluidFlow3D test data. ✓/✗ means our method with/without the zero-divergence loss term. Better results are marked in bold.

| Zero-Divergence | Train Size | EPE | Acc Strict | Acc Relax | Outliers |
|---|---|---|---|---|---|
| ✓ | 100% | **0.0046** | 98.69% | **98.77%** | **1.31%** |
| ✓ | 10% | **0.0061** | **98.27%** | 98.37% | **1.74%** |
| ✓ | 1% | **0.0085** | **97.61%** | **97.76%** | **2.40%** |
| ✗ | 100% | 0.0047 | 98.69% | 98.76% | 1.32% |
| ✗ | 10% | 0.0064 | 98.25% | 98.37% | 1.76% |
| ✗ | 1% | 0.009 | 97.45% | 97.60% | 2.56% |

Table 7: Comparison on different flow cases. ✓/✗ means our method with/without the zero-divergence loss term. Better results are marked in bold.

**Beltrami Flow**

| Zero-Divergence | Train Size | EPE | Acc Strict | Acc Relax | Outliers | NEPE |
|---|---|---|---|---|---|---|
| ✓ | 100% | **0.0039** | **98.95%** | **98.99%** | **1.05%** | **0.0191** |
| ✓ | 10% | **0.0064** | **98.39%** | **98.45%** | **1.62%** | **0.0243** |
| ✓ | 1% | **0.0101** | **97.58%** | **97.66%** | **2.44%** | **0.0319** |
| ✗ | 100% | 0.0040 | 98.93% | 98.97% | 1.08% | 0.0195 |
| ✗ | 10% | 0.0065 | 98.37% | 98.43% | 1.64% | 0.0253 |
| ✗ | 1% | 0.0101 | 97.51% | 97.59% | 2.45% | 0.0319 |

**Turbulent Channel Flow**

| Zero-Divergence | Train Size | EPE | Acc Strict | Acc Relax | Outliers | NEPE |
|---|---|---|---|---|---|---|
| ✓ | 100% | 0.0019 | **99.38%** | **99.41%** | 0.62% | **0.0051** |
| ✓ | 10% | 0.0024 | 99.31% | 99.35% | 0.69% | 0.0063 |
| ✓ | 1% | 0.0025 | **99.25%** | **99.31%** | 0.75% | **0.0065** |
| ✗ | 100% | 0.002 | 99.38% | 99.41% | 0.62% | 0.0052 |
| ✗ | 10% | **0.0022** | **99.33%** | **99.36%** | 0.68% | **0.0059** |
| ✗ | 1% | 0.0028 | 99.21% | 99.26% | 0.79% | 0.007 |

**Forced Isotropic Turbulence**

| Zero-Divergence | Train Size | EPE | Acc Strict | Acc Relax | Outliers | NEPE |
|---|---|---|---|---|---|---|
| ✓ | 100% | **0.0117** | **96.94%** | **97.04%** | **3.07%** | **0.0378** |
| ✓ | 10% | 0.0156 | 96.11% | 96.26% | 3.92% | 0.0515 |
| ✓ | 1% | **0.0193** | 95.01% | 95.22% | 5.02% | **0.0645** |
| ✗ | 100% | 0.0120 | 96.86% | 96.97% | 3.16% | 0.0386 |
| ✗ | 10% | **0.0146** | **96.23%** | **96.39%** | **3.80%** | **0.0482** |
| ✗ | 1% | 0.0193 | **95.08%** | **95.27%** | **4.96%** | 0.0653 |

**Forced MHD Turbulence**

| Zero-Divergence | Train Size | EPE | Acc Strict | Acc Relax | Outliers | NEPE |
|---|---|---|---|---|---|---|
| ✓ | 100% | **0.0037** | **98.89%** | **98.95%** | **1.13%** | **0.0306** |
| ✓ | 10% | 0.0051 | 98.51% | 98.59% | 1.53% | 0.0412 |
| ✓ | 1% | **0.0066** | 97.98% | **98.12%** | 2.06% | **0.0505** |
| ✗ | 100% | 0.0038 | 98.86% | 98.92% | 1.16% | 0.0313 |
| ✗ | 10% | **0.0048** | **98.55%** | **98.64%** | 1.48% | 0.0387 |
| ✗ | 1% | 0.0067 | **97.99%** | 98.12% | **2.05%** | 0.0519 |

**Transitional Boundary Flow**

| Zero-Divergence | Train Size | EPE | Acc Strict | Acc Relax | Outliers | NEPE |
|---|---|---|---|---|---|---|
| ✓ | 100% | 0.0028 | 99.02% | 99.18% | 0.96% | 0.0067 |
| ✓ | 10% | **0.0040** | **98.60%** | **98.81%** | **1.37%** | **0.0093** |
| ✓ | 1% | **0.0066** | **97.55%** | **97.87%** | **2.42%** | **0.0149** |
| ✗ | 100% | 0.0027 | **99.09%** | **99.23%** | **0.89%** | **0.0064** |
| ✗ | 10% | 0.0045 | 98.23% | 98.49% | 1.74% | 0.0103 |
| ✗ | 1% | 0.0089 | 96.57% | 96.91% | 3.40% | 0.0188 |

**Uniform Flow**

| Zero-Divergence | Train Size | EPE | Acc Strict | Acc Relax | Outliers | NEPE |
|---|---|---|---|---|---|---|
| ✓ | 100% | **0.0018** | 99.42% | **99.44%** | 0.58% | 0.004 |
| ✓ | 10% | 0.0020 | **99.40%** | **99.42%** | **0.60%** | 0.0043 |
| ✓ | 1% | 0.0019 | **99.41%** | **99.43%** | 0.59% | 0.0041 |
| ✗ | 100% | 0.0018 | **99.43%** | 99.44% | 0.57% | 0.0039 |
| ✗ | 10% | 0.0020 | 99.40% | 99.42% | 0.60% | **0.0042** |
| ✗ | 1% | 0.0020 | 99.39% | 99.41% | 0.61% | 0.0042 |

Table 8: Comparison on FluidFlow3D test data. ✓/✗ means our method with/without the DVE module. Better results are marked in bold.

| DVE | Train Size | EPE | Acc Strict | Acc Relax | Outliers |
|---|---|---|---|---|---|
| ✓ | 100% | **0.0046** | **98.69%** | **98.77%** | **1.31%** |
| ✓ | 10% | **0.0064** | **98.27%** | **98.37%** | **1.74%** |
| ✓ | 1% | **0.0085** | **97.61%** | **97.76%** | **2.40%** |
| ✗ | 100% | 0.011 | 95.72% | 97.88% | 9.29% |
| ✗ | 10% | 0.016 | 93.38% | 96.93% | 14.59% |
| ✗ | 1% | 0.0219 | 89.92% | 95.56% | 21.04% |

### A.5.3 Dynamic Velocimetry Enhancer(DVE)

We have developed the Dynamic Velocimetry Enhancer (DVE), which is used in the testing phase of the process to optimize the initial flow and can effectively improve cross-domain robustness and data efficiency. To illustrate this, we trained models with and without the DVE module on different sizes of training sets and compared their performance on various test sets. We tested it using the FluidFlow3D test data(See Table. 8) and its six fluid cases(See Table 9).

As evidenced by Tab. 8, the model without the DVE module is less effective than the model with the DVE module, even though the latter is trained on only 1% of the samples. Furthermore, we observe that the model without the DVE module is highly sensitive to the training size, and the performance

Table 9: Comparison on different flow cases. ✓/✗ means our method with/without the DVE module. Better results are marked in bold.

| DVE | Train Size | Beltrami Flow | | | | |
| --- | --- | --- | --- | --- | --- | --- |
| | | EPE | Acc Strict | Acc Relax | Outliers | NEPE |
| ✓ | 100% | **0.0039** | **98.95%** | **98.99%** | **1.05%** | **0.0191** |
| ✓ | 10% | **0.0064** | **98.39%** | **98.45%** | **1.62%** | **0.0243** |
| ✓ | 1% | **0.0101** | **97.58%** | **97.66%** | **2.44%** | **0.0319** |
| ✗ | 100% | 0.0096 | 96.40% | 98.18% | 7.11% | 0.0442 |
| ✗ | 10% | 0.0158 | 93.66% | 96.80% | 14.44% | 0.0709 |
| ✗ | 1% | 0.0230 | 90.38% | 95.10% | 22.63% | 0.0997 |

| DVE | Train Size | Turbulent Channel Flow | | | | |
| --- | --- | --- | --- | --- | --- | --- |
| | | EPE | Acc Strict | Acc Relax | Outliers | NEPE |
| ✓ | 100% | **0.0019** | **99.38%** | **99.41%** | **0.62%** | **0.0051** |
| ✓ | 10% | **0.0024** | **99.31%** | **99.35%** | **0.69%** | **0.0063** |
| ✓ | 1% | **0.0025** | **99.25%** | **99.31%** | **0.75%** | **0.0065** |
| ✗ | 100% | 0.0036 | 99.03% | 99.39% | 1.14% | 0.0094 |
| ✗ | 10% | 0.0056 | 98.57% | 99.22% | 1.86% | 0.0145 |
| ✗ | 1% | 0.0074 | 98.03% | 99.12% | 2.65% | 0.0187 |

| DVE | Train Size | Forced Isotropic Turbulence | | | | |
| --- | --- | --- | --- | --- | --- | --- |
| | | EPE | Acc Strict | Acc Relax | Outliers | NEPE |
| ✓ | 100% | **0.0117** | **96.94%** | **97.04%** | **3.07%** | **0.0378** |
| ✓ | 10% | **0.0156** | **96.11%** | **96.26%** | **3.92%** | **0.0515** |
| ✓ | 1% | **0.0193** | **95.01%** | **95.22%** | **5.02%** | **0.0645** |
| ✗ | 100% | 0.0244 | 89.05% | 94.19% | 19.15% | 0.093 |
| ✗ | 10% | 0.0318 | 85.55% | 92.43% | 26.19% | 0.1267 |
| ✗ | 1% | 0.0396 | 80.94% | 90.18% | 34.46% | 0.1618 |

| DVE | Train Size | Forced MHD Turbulence | | | | |
| --- | --- | --- | --- | --- | --- | --- |
| | | EPE | Acc Strict | Acc Relax | Outliers | NEPE |
| ✓ | 100% | **0.0037** | **98.89%** | **98.95%** | **1.13%** | **0.0306** |
| ✓ | 10% | **0.0051** | **98.51%** | **98.59%** | **1.53%** | **0.0412** |
| ✓ | 1% | **0.0066** | **97.98%** | **98.12%** | **2.06%** | **0.0505** |
| ✗ | 100% | 0.0117 | 95.41% | 98.01% | 19.35% | 0.1032 |
| ✗ | 10% | 0.0169 | 92.66% | 96.97% | 29.36% | 0.1552 |
| ✗ | 1% | 0.0226 | 89.06% | 95.68% | 39.86% | 0.2103 |

| DVE | Train Size | Transitional Boundary Flow | | | | |
| --- | --- | --- | --- | --- | --- | --- |
| | | EPE | Acc Strict | Acc Relax | Outliers | NEPE |
| ✓ | 100% | **0.0028** | **99.02%** | **99.18%** | **0.96%** | **0.0067** |
| ✓ | 10% | **0.0040** | **98.60%** | **98.81%** | **1.37%** | **0.0093** |
| ✓ | 1% | **0.0066** | **97.55%** | **97.87%** | **2.42%** | **0.0149** |
| ✗ | 100% | 0.0082 | 97.42% | 99.03% | 2.68% | 0.0183 |
| ✗ | 10% | 0.0134 | 94.49% | 98.30% | 5.65% | 0.0297 |
| ✗ | 1% | 0.0222 | 88.17% | 96.35% | 12.17% | 0.0494 |

| DVE | Train Size | Uniform Flow | | | | |
| --- | --- | --- | --- | --- | --- | --- |
| | | EPE | Acc Strict | Acc Relax | Outliers | NEPE |
| ✓ | 100% | **0.0018** | **99.42%** | **99.44%** | **0.58%** | **0.004** |
| ✓ | 10% | **0.0020** | **99.40%** | **99.42%** | **0.60%** | **0.0043** |
| ✓ | 1% | **0.0019** | **99.41%** | **99.43%** | **0.59%** | **0.0041** |
| ✗ | 100% | 0.0037 | 99.09% | 99.43% | 0.97% | 0.008 |
| ✗ | 10% | 0.0058 | 98.74% | 99.35% | 1.46% | 0.0124 |
| ✗ | 1% | 0.0074 | 98.38% | 99.33% | 1.92% | 0.0159 |

Table 10: Runtime Profiling of Our Method. Different Train Size settings share the same inference time T_test. The inference process includes the Forward of the trained network and our DVE.

| Train Size | Training Epochs | T_train(h) | T_test(s) | |
| --- | --- | --- | --- | --- |
| 100% | 100 | 16.9 | 0.218 | |
| 10% | 100 | 1.241 | Forward | DVE |
| 1% | 300 | 0.502 | 0.019 | 0.199 |

of each metric rapidly declines as the training size decreases. In contrast, the model with the DVE module exhibits a certain degree of robustness. This demonstrates the significant impact of our DVE module on data efficiency.

Tab. 9 shows the performance of the DVE module also varies across different flow cases. In simple cases, the presence or absence of the DVE module has minimal impact on the metrics, since the initial flow already closely matches the ground truth with a relatively small error. However, in challenging cases where the initial flow is far from the ground truth, the DVE module plays a crucial role. It significantly enhances the model's accuracy through a straightforward and focused optimization process.

**Time Profiling of DVE**   A significant challenge in test-time optimization is implementing per-sample adaptation at test time without adversely affecting inference efficiency. We demonstrate the time profiling of our method in Table 10 to illustrate the efficiency of our proposed test-time optimization paradigm. Additionally, we present the convergence graph of DVE to study the influence of the number of refinement steps. As shown in Figure 9, DVE can converge within 150 epochs due to its simple structure, resulting in fast inference and high efficiency.

## A.6   Limitations

Our research addresses the challenge of estimating dual-frame fluid motion. However, this field suffers from a scarcity of large-scale benchmark datasets. To mitigate this issue, we incorporate real-

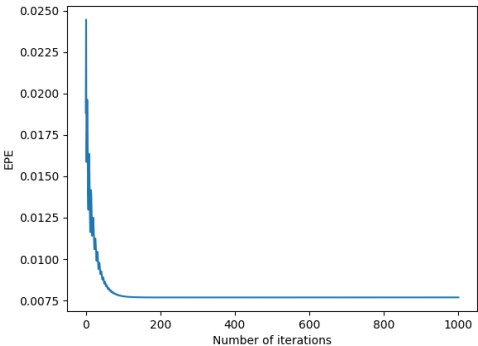

Figure 9: Convergence graph of EPE with respect to the number of refinement steps (iterations).

world datasets and conduct extensive studies across diverse flow cases. Nonetheless, it is anticipated that a broader variety of fluid motion data will become available in the future.

### A.7 Impact Statements

Our research focuses on advancing a dual-frame fluid motion estimation method designed for optimal data efficiency and cross-domain robustness in turbulent flow analysis. This innovative approach is anticipated to empower scientists to analyze fluid dynamics with significantly reduced data requirements while enhancing overall method robustness. It is crucial to acknowledge potential limitations, particularly in real-world applications like blood analysis in medical science, where errors arising from our method could potentially be harmful. Our intent is to continually refine and improve our methodology to minimize any such unintended consequences.

