# OpenReview forum: "Dual-frame Fluid Motion Estimation with Test-time Optimization and Zero-divergence Loss"
_NeurIPS.cc/2024/Conference — NeurIPS 2024 poster_

### Official Review · Reviewer_N6ux · 2024-07-10

**Soundness:** 2
**Presentation:** 2
**Contribution:** 2
**Rating:** 5
**Confidence:** 4

**Summary:**

In this paper the authors propose a graph based network that combines feature extraction with test time optimization to perform two-frame particle tracing. The core problem is that given two sets of points, e.g., point cloud data, to find the correspondences between the point clouds and estimating their velocity from this correspondence. While this problem can be well analyzed in synthetic data, real world problems are often sparse in data and, accordingly, using methods that do not require labeled data (and not a lot of it) would be beneficial. The proposed methodology works by proposing an unsupervised loss formulation based on assuming smoothness and divergence-freedom of the solution space and uses parts of this loss function as a test-time target for optimization. The method is evaluated and trained on synthetic data and then evaluated on dissimilar data to show the generalization capabilities of the approach.

**Strengths:**

The overall idea and motivation of the paper is useful and would solve an important problem. The strengths of the paper are:
* A method that can train with significantly smaller datasets than current state of the art methods at better performance
* Test time optimization for particle tracking
* A physically motivated loss term to drive a self-supervised learning setup
* The proposed method is, fairly, straightforward and works in a variety of settings with one-out training

**Weaknesses:**

Overall the paper has an interesting core idea, however, several issues exist, especially regarding evaluation and the lack of statistical evaluations and ablation studies in general. There are a few core weaknesses listed below in a short list. Following these a list of all potential issues and weaknesses is included for completeness at this point. Note that many of these issues are also addressed separately and are not questions in themselves.
* The authors perform no evaluation regarding the statistical significance of their proposed method. While the computations are deterministic (as they note in the checklist), this does not mean that network initialization does not play a role, especially when many of the evaluation metrics are relatively close and some results seem counter intuitive, e.g., the EPE improving for smaller datasets. Furthermore, there is no clear evaluation on the choice of the 1% of data, i.e., did the authors try different subsets? Were the 1% chosen consistently across class? Across different trained methods?, etc.
* It is not clear why the proposed DVE test-time optimization cannot be performed on top of data trained on supervised data, especially as the authors change the loss functions anyways from training to testing.
* The authors make a fairly broad claim in the introduction as to synthetic data being always limited due to hand-crafted priors but show that their method does not suffer from this problem, even though it uses the same synthetic data. This needs to be clarified.
* The divergence-free loss term is not very well evaluated. On the one hand the assumption seems to be relatively central to the approach and the authors go out of their way to state that assuming divergence-freedom can even be assumed in compressible cases (where it does not hold), but then also state that they disable the divergence-free term during test-time optimization especially for cases where it does not hold. * Either the authors need to more clearly evaluate the impact of the divergence-free loss term during test-time optimization in _all_ cases, or consider the importance of the argument.
* The  formatting of the paper at times is very odd with subsubsubsubsections and randomly highlighting of statements with inconsistent (and incomplete) highlighting in tables. This should be improved.

General:
* In the paper checklist under 7 the authors note that their ‘method is deterministic’ and thus does not require error bars. However, choosing random subsets of data, training different architectures and more usually involves sources of randomness and seed variance can be quite significant. This needs to be evaluated properly and adjusted accordingly.
* Table 3 seems odd as the EPE _improves_ for the proposed method when training on smaller datasets.
* Table 9: Why is the Epoch count increased to 300 for the 1% train size and is this done consistently across all methods and what would the results be with only 100 Epochs or a normalized training with the same number of weight updates across dataset sizes?
* It is wholly unclear as to why the proposed scheme is 6 times slower during testing just because it is trained on a larger dataset.
* Why can other methods not utilize DVE as a test-time optimization process? This would make for much fairer comparisons
* The authors should make it more clear in the related work as to which graph-based feature extractor they use as their architecture is heavily informed by prior work.

Dataset:
* The authors claim that synthetic data is always limited as it leads to ‘hand-crafted priors’ but they utilize a synthetic dataset to train their method and show that it is not limited by the biases induced from the prior, i.e., their method can generalize even from synthetic data.
* The authors state that in cases of boundary conditions, or other situations, smoothness and divergence-freedom is not holding up. This should be more clearly shown, i.e., a demonstration that using  these regularization terms leads to issues in such cases.

Reduced Data Training:
* The 1% setup is interesting, however, the authors do not clearly show how sensitive their method (or methods in general) are to selection, i.e., there is no clear evaluation on how the choice of the 1% affects the outcome
* There are no statistical evaluations, which would be especially important for the 1% evaluations and with the difference between methods being so narrow at times, initial seed choice might be important too
* There is no evaluation as to what the method would do if the data is incomplete, i.e., if particles are missing from one of the frames. This would be important for real-world applications but due to their reliance on hand-crafted synthetic priors and other narrow datasets, this is not evaluated clearly.

Divergence-Freedom/Loss Terms:
* The Divergence-free loss term is interesting but not using it during testing seems odd as it should be helpful in all cases. However, even the authors note that this would be a limitation in non divergence free settings (which they evaluate one of). Considering the prominence of the divergence-free claim, this needs to be further elucidated (especially considering the statement in line 674)
* The authors claim (178) that even for compressible fluids divergence-freedom can be assumed in practice. While sometimes done, this should be more clearly evaluated as the authors clearly do not use the divergence-free loss in datasets where the assumption would not hold.
* The distinctions when smoothness and divergence-freedom are not to be used for optimization are not clear as there might be reason to not use the divergence-free loss, i.e., if one expects a dataset with divergence, but not requiring smoothness is odd. This should be done in an ablation study.
* The neighborhood sizes are odd and should be more clearly written as the information in 715 is not easily understood, i.e., is the divergence-loss computed with only 2 points?
* The formulation of the divergence as a central finite difference scheme is not necessary, other information would be more prudent.

Formatting/Writing:
* The formatting is very inconsistent at times, e.g., table layouts, subsubsubsubsection headings are sometimes italicized, sometimes underlined and sometimes bold.
* The authors do not need to state that they are excluding information due to page limits, twice.
* The highlighting in Figure (mostly table) 4 is inconsistent. In the top part the second best performing method is not highlighted with underlining but also boldfaced and the T_test column has no highlighting
* Table 6 should also highlight cases where the method without divergence-freedom performs better

Missing Details:
* How were the lambda parameters chosen?
* KNN should be properly introduced (208) and evaluated as to why it is not efficient

Evaluation:
* The method performs significantly worse in some cases, e.g., the EPE is worse by a factor of 2 for the MHD case compared to the turbulent channel flow. It would be nice if this was more clearly evaluated and highlighted, especially for the Beltrami case (Figure/Table 5) where Gotflow performs worse by a factor of 2 relative to the proposed method, which is fairly close considering this case is about 8 times worse than the best case for the proposed method and only 3 times worse for Gotflow.
* The improvement of 22882 matches over 22001 seems fairly minor (315) and it would be prudent to add the tracking timing in this section in the main paper.
* How does the proposed method perform on its own (323), i.e., when it is not used as an initializer?
* The difference in EPE from 100% to 10% with and without DVE is relatively close even though the text (800) notes that the method is highly sensitive.

**Questions:**

Note that these questions are not in order of importance.
1. What is the variance and statistical certainty of the training across different initializations for the proposed methods?
2. Why is the EPE error lower for the 1% case in Table 3?
3. What would the training results be with equal weight updates across methods, and if they do not change from the proposed hyperparameters, why were these hyperparameters chosen?
4. Is it possible to apply DVE to other state of the art methods and, if so, what would the results be?
5. How was the 1% subset of the data chosen? Does this influence the result, i.e., do different subsamplings (and strategies) result in consistent (within a small deviation) results?
6. Is there any data to highlight the limitations of including the regularization terms during test time optimization?
7. How were the lambda parameters chosen?
8. Why is the computational time of the method at test time different based on the dataset size used during training?
9. What was the total time to train a network from start to finish for the proposed method with and without the various terms?

**Limitations:**

The primary limitations of the paper relate to the evaluation of the methodology and the divergence-free loss term. In summary there are three key issues:
1. The evaluation should include a broader evaluation that at least considers seed initialization and variance due to the 1% sampling. As the paper currently stands, it is impossible to tell, with certainty, that the results are consistent and not just due to a lucky seed. While performing such evaluations for all setups is computationally prohibitive, at least the core contributions should be evaluated
2. There is a lot of inconsistency regarding the divergence-free loss. On one hand it is argued as very important during training and how it can work even in compressible cases. Then the term is wholly disabled during test time optimization because it isn’t needed and then towards the end in a case with divergence the term is left off anyways. There is no clear evaluation as to the influence of including the divergence-free condition during test optimization, especially for cases which are compressible (and/or not divergence-free). This makes it difficult to judge how the method truly generalizes
3. While not a direct limitation, the authors themselves state that the use of synthetic data leads to hand-crafted biases (in the introduction), but they also demonstrate how their method is immune to this. As this does not seem to be a limitation of the synthetic data nature, this should be either clarified in the introduction or an actual limitation where the method does not generalize due to its limitation from hand-crafted data.
Rating:

---

> ### Author Rebuttal · Authors · 2024-08-07
>
> ### Q1: 1% chosen consistently across class?
> **Answer:**
> Yes, the sub-sampled dataset is chosen consistently.
> ### Q2: Statistical Significance
> **Answer:**
> We control experiment randomness with seeds, conducting multiple runs to ensure variability while maintaining consistency with the same seed. Results in Tables 2 and 3 of the rebuttal PDF affirm our method's robustness.
> ### Q3: DVE performed on top of data trained on supervised data？
> **Answer:** DVE integration during training isn't feasible as gradients cannot propagate through the DVE module and it's also against our training objective.
> ### Q4: Use synthetic data yet effectively mitigates its limitations.
> **Answer:**
> We train solely with synthetic data, aiming to develop a robust feature extractor that recognizes basic correspondences consistent across synthetic and real data. Our strategy maximizes the advantages of synthetic data and addresses domain shifts at test time. Our method effectively mitigates these shifts using the DVE module during test-time optimization, as explained in Section 4.3 of our main paper.
> ### Q5: Why to disabe zero-divergence during DVE.
> **Answer:** We deactivate the zero-divergence loss during test time when it may not be precise, as inaccuracies in loss terms can negatively impact optimization. Specifically, zero-divergence only holds when the fluid is incompressible—true for our evaluation scenarios—and the flow field is sufficiently dense to compute divergence accurately, which isn't always the case in our tests.
> ### Q6: Evaluation of the Zero-Divergence/Smooth loss term during DVE
> **Answer:**
> We argue against using regularizers like zero-divergence or smooth loss during test-time optimization if they might reduce accuracy. Our studies, detailed in Table 4 of our rebuttal PDF, show that smooth loss generally degrades performance, and zero-divergence loss only improves it in specific scenarios like Transition. This is because zero-divergence applies effectively only in incompressible and sufficiently dense flow fields, conditions not consistently met in our tests.
> ### Q7: Formatting Issues:
> **Answer:** We'll fix the writing issues and adjust the formatting in the revised version.
> ### Q8: Odd Results:
> **Answer:**
> - Figure 4 upper table: T_test for "Ours" should be 0.218s rather than 1.353s.
> - Table 3: Ours should be as Table 5 in rebuttal PDF.
> ### Q9: Epoch increased to 300 for the 1% train size
> **Answer:** It's because the 1% training size setting requires a longer time to converge.
> ### Q10: Training with the same epoch num across dataset sizes:
> **Answer:** We trained our method with 300 and 100 epochs across dataset sizes, as shown in Table 6 of the rebuttal PDF. The results indicate that 100 epochs are insufficient for convergence when training on only 1% of the data.
> ### Q11: Why our method is 6 times slower when trained on a larger dataset.
> **Answer:**  Please see Q8.
> ### Q12: Other methods with DVE for much fairer comparisons.
> **Answer:**  Integrating DVE into other methods wouldn't ensure a fair comparison as it is essentially comparing our basic feature extractor against more complex systems.
> Results in Table 7 of the rebuttal PDF show that adding the DVE module can significantly improve the performance. Although GotFlow3D w/ DVE performs better, its network is time-consuming.
> ### Q13: Lack of graph-based feature extractor in the related works:
> **Answer:**   We have introduced its related works in Section 3.2.2 of the main paper. We will revise the related works.
> ### Q14: Missing particles from frames:
> **Answer:**
> To simulate missing particle scenarios, we downsample particles in either the source or target frame. The results are detailed in Table 8 of the rebuttal PDF.
> ### Q15: Not using zero-divergence during testing.
> **Answer:**  Assuming it works for all cases is not the best practice. Please refer to Q5 and Q6 for details.
> ### Q16: Neighborhood sizes
> **Answer:**  It is not sensitive. See Table 9 of the rebuttal PDF.
> ### Q17: Use smooth loss for DVE
> **Answer:**  We argue that smooth loss is only a regularizer and we should not assume it holds for all datasets. Please refer to Q5 and Q6 for more details.
> ### Q18: Need other information about divergence calculation
> **Answer:**  We do not understand what "other information" refers to here. For our implementation, we use torch.gradient.
> ### Q19: Lambda chosen
> **Answer:** By cross validation.
> ### Q20: Evaluation on KNN.
> **Answer:** Direct KNN application is ineffective in sparse fields. Our method as a grid-based gradient computation is more precise due to the well-established techniques in gradient calculations on grids. A  comparison is shown in Table 10 of the rebuttal PDF.
> ### Q21: More clearly evaluated why the method performs significantly worse in some cases.
> **Answer:**
> We provide visualizations of different flow cases in rebuttal PDF. Our method excels in simpler flow cases like "Channel," "Transition," and "Uniform," but performs less better in complex scenarios due to inherent limitations.
> ### Q22: Prudent to add the tracking timing (Line 315)
> **Answer:** It is already included in Figure 6-c. We will revise it further.
> ### Q23: Method performance on its own when not used as an initializer
> **Answer:** Refer to Q1 of our reply to jdpi.
> ### Q24: Performance too close in DVE ablation study
> **Answer:**  We argue that the difference is significant. EPE has a notable scale difference (e.g., from 0.0046 to 0.011 at 100% and from 0.0064 to 0.016 at 10%), and other metrics, such as the outlier rate, are also significant.
> ### Q25: Hyperparamter chosen for baselines
> **Answer:**  Default hyperparameters ensure optimal performance, with further experimentation shown in Table 11 of the rebuttal PDF where GotFlow3D overfits at 100 epochs.
> ### Q26: Apply DVE to other methods
> **Answer:**  Please refer to Q12.
> ### Q27: Training time
> **Answer:**  See Table 9 of main paper. It should be roughly the same without loss terms.

---

> > ### Comment · Area_Chair_Eatq · 2024-08-09
> > **Request from AC**
> >
> > Dear Reviewer N6ux,
> >
> > Thank you very much for your detailed review comments.
> > The authors carefully provided comments in the rebuttal. Would you please reply to them, if needed?
> >
> > Best,

---

### Official Review · Reviewer_wM8x · 2024-07-14

**Soundness:** 3
**Presentation:** 3
**Contribution:** 3
**Rating:** 6
**Confidence:** 3

**Summary:**

The paper proposes a self-supervised method to learn 3D particle tracking and modelling turbulent fluid flow. They regularize their method with a zero-divergence loss function and inspired by the splat operation they propose a splat-based implementation for this loss. They also incorporate a GCNN feature extractor to learn geometric features. Their method also supports test-time optimization, with their Dynamic Velocimetry Enhancer (DVE) module.

**Strengths:**

Strengths
The paper writing is good.
Novelty: the paper suggested a novel approach to solving the large data dependency problem. I really liked the idea of using EdgeConv to incorporate geometric information.
The paper tackles an important problem.

**Weaknesses:**

* The method needs test-time optimisation, which will make the result a bit questionable if you optimize the test set.
* The test time complexity is not reported clearly. In Fig 4 (top table, ours (1%) w/o DEV) it is not clear whether the time reported is the time for the forward pass or includes the optimization.

**Suggestion:**
Please use single tables format their different formats in the main text and appendices. Also, splitting the tables and figures will make it easier to follow the paper and understand the result.
And remove the unnecessary **bold** used in the main text.

**Questions:**

In Fig 4 (top table, ours (1%) w/o DEV) it is not clear whether the time reported is the time for the forward pass or includes the optimization, can you comment on that?

**Limitations:**

Yes.

---

> ### Author Rebuttal · Authors · 2024-08-07
>
> ### Q1: Test-time optimisation will make the result a bit questionable
>
> **Answer:**
> We note that the test-time optimization is self-supervised without accessing ground truth labels, making the setting realistic. This approach has garnered significant attention recently, as seen in references introduced in the related works (Section 2.1) of our main paper.
>
> Additionally, to make comparisons with baselines fair, we incorporate the DVE test-time optimization module with other baseline methods to ensure a fair comparison.
>
> Due to computing resource constraints, we selected two baselines for comparison: the state-of-the-art method GotFlow3D and FLOT. Both methods include an optimization module in their network and learn to optimize. We integrate our DVE module with their estimated outputs, iterating for the same number of steps as our method. The results are shown below:
>
> |Methods    |T_test	|EPE	|Acc Strict|	Acc Relax|	Outliers
> |----------|----------|----------|----------|----------|----------|
> |Ours|	0.218s|	0.0046	|98.69%|	98.77%|	1.31%|
> |GotFlow3D|	0.758s	|0.0049	|93.15%|	96.38%|	3.62%|
> |GotFlow3D w/ DVE|	2.260s|	0.0024|	99.12%|	99.13%|	0.86%|
> |FLOT	|0.030s	|0.0587	|24.99%|	45.59%|	54.41%|
> |FLOT w/ DVE|	0.520s|	0.0300	|90.38%	|91.15%|	10.00%|
>
> It can be seen that adding the DVE module can significantly improve the performance of GotFlow3D and FLOT, demonstrating that the DVE module is generalizable to other methods. Although GotFlow3D with the DVE module performs better than our method on various metrics, its network is heavy and time-consuming.
>
> ### Q2: In Fig 4 (top table, ours (1%) w/o DVE) it is not clear whether the time reported is the time for the forward pass or includes the optimization.
>
> **Answer:**
> To clarify, The forward pass takes 0.019s and the test-time optimization module takes 0.199s, adding up to 0.218s. This has been stated in Table.9 of the supplementary material.
>
> As for the T_{test} number in Fig.4-Top:
>
> Ours(1%) includes both forward pass and test-time optimization, so the time is 0.218s.
>
> Ours(1%) w/o Div Loss includes both forward pass and test-time optimization, so the time is 0.218s.
>
> Ours(1%) w/o DVE means the test-time optimization module is not included, so the time is 0.019s.
>
> ### Q3: Suggestion: Please use single tables format their different formats in the main text and appendices. Also, splitting the tables and figures will make it easier to follow the paper and understand the result. And remove the unnecessary bold used in the main text.
>
> **Answer:**
> Thanks. We will re-formatted the manuscript according to these points to improve the readability. We'll change all tables to three-line, header bolded. Please see Table 1 in the global rebuttal pdf as an example.

---

> > ### Comment · Area_Chair_Eatq · 2024-08-09
> > **Request from AC**
> >
> > Dear Reviewer wM8x,
> >
> > Thank you very much for your detailed review comments.
> > The authors carefully provided comments in the rebuttal. Would you please reply to them, if needed?
> >
> > Best,

---

### Official Review · Reviewer_jdpi · 2024-07-15

**Soundness:** 3
**Presentation:** 3
**Contribution:** 3
**Rating:** 6
**Confidence:** 4

**Summary:**

In this paper, the author proposes a self-supervised learning based 3D particle tracking velocimetry (PTV) technique for dual-frame fluid motion estimation. The proposed method surpasses its supervised counterparts while utilizing only 1% of the training samples (without labels) compared to previous methods. A zero-divergence loss is utilized for turbulence flow and it is implemented in splat-based approach for efficiency and effectiveness. On the benchmarks, the proposed method shows the best performance comparing to baselines.

**Strengths:**

- The self-supervised learning based method can achieve better performance given only 1% of training samples compared to the previous supervised methods.
- The proposed method shows the best performance overall on benchmarks especially on complex case such as Forced MHD Turbulence.
- The paper is well organized and easy to follow.

**Weaknesses:**

- As for tests on the real-world dataset, the author mentioned the proposed method is integrated with the PTV framework. Just wondering how will the proposed method perform stand alone?
- As mentioned in the paper, there are datasets with sparse flow field, which may influence the performance of the proposed method. It seems that there is no investigation of the particle density. Just wondering how will the method perform given trained on sparse/dense and inference on dense/sparse?

**Questions:**

See weaknesses.

**Limitations:**

Yes. The author discusses the limitations in the appendix.

---

> ### Author Rebuttal · Authors · 2024-08-07
>
> ### Q1: How will the proposed method perform stand alone compared with being integrated into PTV?
>
> **Answer:**
> We cannot perform this evaluation because the task of PTV is a superset of dual-frame motion estimation. Here, we clarify these differences:
>
> Our method targets particle motion vector estimation between two frames, whereas PTV tracks the movement of particles across the whole sequence. The relationship is similar to that between two-frame optical flow (e.g., RAFT [B]) and long-term point racking (e.g., OmniMotion [A]). The PTV framework [C] involves particle detection, particle positioning, dual-frame motion estimation and whole-sequence particle tracking. While our method provides good dual-frame motion initialization, it does not perform whole-sequence matching. Thus, it is not feasible to evaluate our method independently on PTV tasks.
>
> [A] Tracking everything everywhere all at once, ICCV 2023
>
> [B] Raft: Recurrent all-pairs field transforms for optical flow, ECCV 2020
>
> [C] 2-dimensional particle tracking velocimetry (ptv): technique and image processing algorithms. Experiments in fluids, 6(6):373–380, 1988.
>
> To summarize, the key differences are:
>
> - PTV tracks particles across the whole sequence, while our method deals with frame pairs.
> - Dual frame motion estimation methods (like ours and GotFlow) output motion vectors while PTV methods associate particles in a long sequence.
>
> ### Q2: The method performance given trained on sparse/dense and inference on dense/sparse?
>
> **Answer:**
> First, we note that our data originates from physical phenomena (simulation or real-world capture). Manually downsampling it might not accurately reflect the sparse distribution of fluid particles in a real-world scenario.
>
> Second, we provide experiments where models are trained on dense and tested on sparse particle distributions. The particle density is controlled by downsampling particles in each sample. And for fair comparison, we evaluate the model trained on 1% samples of the whole dataset, which corresponds to the main experiment Ours(1%) row in Fig.4-Top.
>
> Third, we demonstrate our model trained on dense particles (100% original particles) and tested on particles with varying levels of sparsity (sample ratios).
>
> | Down-sample Ratio | EPE| 	Acc Strict|	Acc Relax | Outliers |
> |----------|----------|----------|----------|----------|
> |100%| 0.0085|	97.61%|	97.76%|	2.40%|
> | 99%    | 0.0128   | 96.64% | 96.78% | 3.39%  |
> | 95%    | 0.0347   | 91.19%|91.49%|8.84%   |
> | 90%    | 0.0669|83.99%|84.46%|16.04% |
> | 80%    | 0.1363   | 69.91%|70.59%|30.14%   |
> | 50%    | 0.2842|41.02%|	41.86%|	59.03%|
>
> It is important to note that this naive downsampling can cause many particles to lose their corresponding targets, adding substantial difficulty to flow learning.

---

> > ### Comment · Area_Chair_Eatq · 2024-08-09
> > **Request from AC**
> >
> > Dear Reviewer jdpi,
> >
> > Thank you very much for your detailed review comments.
> > The authors carefully provided comments in the rebuttal. Would you please reply to them, if needed?
> >
> > Best,

---

> > ### Comment · Reviewer_jdpi · 2024-08-11
> > **Response to the reply.**
> >
> > Thanks for the detailed replies to my concerns. I don't have further questions and I remain positive about this paper.

---

### Author Rebuttal · Authors · 2024-08-07

We present all tables and figures mentioned in the rebuttal.

---

### Decision · Program_Chairs · 2024-09-25

**Decision:**

Accept (poster)

**Comment:**

All review scores are positive: WA, WA, and BA. While no A and SA are given, AC confirmed that most reviewers' comments are positive. AC also verified that their final evaluations lean toward accepting this paper. Reviewers jdpi and N6ux recognize that the proposed method can train with significantly smaller datasets than current state-of-the-art methods at better performance. Reviewers jdpi and wM8x appreciate that the paper is well organized. Reviewers wM8x and N6ux agree that this work tackles an important and interesting research problem. Finally, reviewers jdpi and N6ux recognize that the experimental results demonstrate the effectiveness of the proposed method. While reviewer N6ux was initially a bit underwhelmed by this paper, the authors' rebuttal did a good job explaining the issues to convince this reviewer. For the reasons stated above, AC recommends this paper for presentation in NeurIPS. Since several issues remain, we expect the authors to revise the manuscript based on the reviewers' comments.